EMBO
reports

# Chronic cold exposure enhances glucose oxidation in brown adipose tissue

Zhichao Wang[1,†] (ID), Tinglu Ning[1,†] (ID), Anying Song[1], Jared Rutter[2,3] (ID), Qiong A Wang[1,4,*] (ID) &
Lei Jiang[1,4,**] (ID)

## Abstract

The cultured brown adipocytes can oxidize glucose *in vitro*, but it is still not fully clear whether brown adipose tissue (BAT) could completely oxidize glucose *in vivo*. Although positron emission tomography (PET) with [18]F-fluorodeoxyglucose ([18]F-FDG) showed a high level of glucose uptake in the activated BAT, the non-metabolizable [18]F-FDG cannot fully demonstrate intracellular glucose metabolism. Through *in vivo* [U-[13]C]glucose tracing, here we show that chronic cold exposure dramatically activates glucose oxidation in BAT and the browning/beiging subcutaneous white adipose tissue (sWAT). Specifically, chronic cold exposure enhances glucose flux into the mitochondrial TCA cycle. Metabolic flux analysis models that β3-adrenergic receptor (β3-AR) agonist significantly enhances the flux of mitochondrial pyruvate uptake through mitochondrial pyruvate carrier (MPC) in the differentiated primary brown adipocytes. Furthermore, *in vivo* MPC inhibition blocks cold-induced glucose oxidation and impairs body temperature maintenance in mice. Together, mitochondrial pyruvate uptake and oxidation serve an important energy source in the chronic cold exposure activated BAT and beige adipose tissue, which supports a role for glucose oxidation in brown fat thermogenesis.

**Keywords** BAT; *in vivo* glucose tracing; metabolic flux analysis; mitochondrial pyruvate carrier

**Subject Category** Metabolism

## Introduction

Upon cold exposure, brown adipose tissue (BAT) is specialized in dissipating chemical energy in the form of heat through adaptive nonshivering thermogenesis, primarily through uncoupled mitochondrial respiration (Chouchani *et al*, 2016, 2019). Besides BAT, chronic cold exposure also induces browning/beiging of the subcutaneous white adipose tissue (sWAT). Earlier studies, including ours, show that the newly generated beige adipocytes have similar morphology and function to the brown adipocytes, and the beige adipocyte-enriched sWAT is also known as beige adipose tissue (Wu *et al*, 2012; Wang *et al*, 2013, 2015). As thermogenic BAT and beige adipose tissue can function as energy sink, they play a key role in regulating glucose and lipid homeostasis as well as whole body insulin sensitivity (Stanford *et al*, 2013; Hanssen *et al*, 2015).

Positron emission tomography (PET) with the tracer [18]F-fluorodeoxyglucose ([18]F-FDG) is widely used to image brown/beige adipose tissue in humans and rodents based on the high level of glucose uptake in the BAT and beige adipose tissue (Cypess *et al*, 2009; Lichtenbelt *et al*, 2009; Virtanen *et al*, 2009; Zhang *et al*, 2018). It is worth noting that [18]F-FDG-PET was originally developed to detect the enhanced glucose uptake and retention in tumors, since most cancer cells exhibit dramatically elevated glucose uptake and lactate secretion, also known as Warburg Effect (Schworer *et al*, 2019). However, the accumulation of [18]F-FDG, a non-metabolizable analogue of glucose, only shows glucose uptake. Different from [18]F-FDG, [U-[13]C]glucose is a fully metabolizable tracer, and the recent *in vivo* application of [U-[13]C]glucose tracing showed that, in addition to Warburg Effect, glucose oxidation is fully active in the tumor of patients with lung cancer (Hensley *et al*, 2016; Faubert *et al*, 2017).

Since [18]F-FDG-PET only shows the activated glucose uptake in the brown/beige adipose tissue, PET imaging with an [11]C-acetate tracer is used to show the enhanced mitochondrial oxidative metabolism in thermogenic brown/beige adipose tissue (Ouellet *et al*, 2012; Blondin *et al*, 2015). However, it is not fully clearly whether glucose is metabolized into acetyl-CoA in the brown/beige adipose tissue *in vivo*. Although *in vitro* studies showed that glucose oxidation can be stimulated in cultured brown adipocyte cell lines (Shackney & Joel, 1966; Isler *et al*, 1987; Irshad *et al*, 2017; Held *et al*, 2018), it is still not fully clear whether the thermogenic brown/beige adipose tissue could completely oxidize glucose *in vivo* (Ma & Foster, 1986; Hankir & Klingenspor, 2018; Fischer *et al*, 2020).

1 Department of Molecular & Cellular Endocrinology, Diabetes and Metabolism Research Institute, City of Hope Medical Center, Duarte, CA, USA
2 Howard Hughes Medical Institute, University of Utah School of Medicine, Salt Lake City, UT, USA
3 Department of Biochemistry, University of Utah School of Medicine, Salt Lake City, UT, USA
4 Comprehensive Cancer Center, Beckman Research Institute, City of Hope Medical Center, Duarte, CA, USA
*Corresponding author. Tel: +1 626 218 6419; E-mail: qwang@coh.org
**Corresponding author. Tel: +1 626 218 6401; E-mail: ljiang@coh.org
†These authors contributed equally to this work

Through *in vivo* [U-$^{13}$C]glucose tracing, here we showed that 10-days chronic cold exposure dramatically activated glucose oxidation in mouse BAT and sWAT. We also found that, comparing to the undifferentiated stromal vascular fractions (SVF), the differentiated primary brown adipocytes were more oxidative and less glycolytic. Furthermore, β3-adrenergic receptor (β3-AR) agonist significantly enhanced glucose oxidation by elevating the flux of mitochondrial pyruvate carrier (MPC), which connects glucose-dependent cytosolic and mitochondrial metabolism (Bricker *et al*, 2012; Herzig *et al*, 2012). This β3-AR-stimulated MPC flux in primary brown adipocytes was consistent with a recent *in vitro* study, which showed the enhanced pyruvate dehydrogenase (PDH) in the differentiated T37i cell line (Held *et al*, 2018). Importantly, the chemical inhibition of MPC blocked the enhanced glucose oxidation in both *in vivo* and *in vitro* models, and *in vivo* MPC inhibition impaired body temperature maintenance in the cold-exposed mice. Together, these data indicate that MPC-mediated glucose oxidation is an important energy source in the chronic cold exposure activated BAT.

## Results

### Chronic cold exposure induced oxidative metabolism in brown adipose tissue (BAT)

To reveal how glucose is metabolized in the BAT of mice upon chronic cold exposure, a metabolizable [U-$^{13}$C]glucose tracer was intraperitoneally (IP) injected into mice individually housed at 6°C (cold exposure) or 30°C (thermoneutral condition for mice) for 10 days. 15 minutes after bolus injection of the [U-$^{13}$C]glucose tracer, over 40% of the circulating and intra-BAT glucose was enriched as m+6 glucose (containing all 6 $^{13}$Carbons from the [U-$^{13}$C]glucose tracer), and the glucose enrichment in serum and BAT was lower in mice housed at 6°C (Fig 1A). Surprisingly, although serum glucose concentration was similar at cold and thermoneutrality, the glucose level in the BAT of mice housed at 6°C was < 10% of that of male mice housed at 30°C (Fig 1B). While mice housed at 30°C contained a wide range of BAT glucose concentrations, all thermoneutral mice showed higher intra-BAT glucose than the mice housed at 6°C (Fig 1B). The low level of [U-$^{13}$C]glucose may be reflective of rapid glucose consumption in the BAT of mice housed in 6°C that outpaces the rate of glucose delivery. This result is not controversial with the robust PET imaging signal of $^{18}$F-FDG in the cold-exposed BAT, as $^{18}$F-FDG can be accumulated upon uptake.

In most mammalian cells, glucose is metabolized to pyruvate through the cytosolic glycolysis, and pyruvate is further oxidized to $CO_2$ through the mitochondrial TCA cycle. Per one molecule of glucose, the cytosolic glycolysis produces 2 molecules of ATP and 2 molecules of NADH, and the mitochondrial TCA cycle produces 2 molecules of ATP, 8 molecules of NADH, and 2 molecules of FADH$_2$. As NADH and FADH can be further metabolized to produce much more ATP, one molecule of glucose produces 7 molecules of ATP through glycolysis, and 25 molecules of ATP through TCA cycle. The carbon transitions of these steps of [U-$^{13}$C]glucose metabolism are illustrated in Fig 1C. For simplicity, only the major metabolite isotopologues are illustrated and presented in Fig 1, with the full enrichment of the metabolites in BAT being displayed in Appendix Fig S1. Previous FDG-PET studies showed that glucose

uptake was increased in both BAT and tumor, and earlier *in vivo* [U-$^{13}$C]glucose tracing confirmed the activated glycolysis in human and mouse lung tumors (Faubert *et al*, 2017). Surprisingly, our *in vivo* [U-$^{13}$C]glucose tracing suggests that chronic cold exposure did not alter the enrichment of glycolytic intermediates in BAT. The m+3 enrichments of glycolytic intermediates, 3-phosphoglycerate (3PG), phosphoenolpyruvate (PEP), pyruvate, and lactate, were unchanged in BAT following chronic cold exposure (Fig 1D). The unchanged 3PG enrichment from [U-$^{13}$C]glucose tracer was also observed in BAT of mice upon acute cold exposure (Mills *et al*, 2018).

In addition to glycolysis, the mitochondrial TCA cycle and oxidative phosphorylation comprise another major energy-producing pathway. Enhanced oxidative metabolism in BAT upon cold exposure was observed by $^{11}$C-acetate and $^{18}$F-fluoro-thiaheptadecanoic acid ($^{18}$FTHA) tracing in both rodent models and humans (Ouellet *et al*, 2012; Blondin *et al*, 2014, 2015; Labbe *et al*, 2015). Our *in vivo* [U-$^{13}$C]glucose tracing directly showed that chronic cold exposure significantly increased the m+2 enrichment of TCA cycle intermediates (citrate, αKG, succinate, fumarate, and malate) in BAT of male mice (Fig 1E). The relative m+2 enrichment of TCA cycle intermediates was induced by 25–59%, if data are normalized to the average enrichment of each metabolite in the 30°C group (Fig EV1A). The increased m+2 enrichment of TCA cycle intermediates reflected the increased flux of glucose oxidation, as the relative abundance of TCA cycle intermediates in BAT was not altered by cold exposure, except for the significant induction of succinate (Fig EV1B). The induced succinate level in BAT is consistent with a recent study, which showed the induced succinate in BAT after 3 h of acute cold exposure (Mills *et al*, 2018). Thus, chronic cold exposure enhances mitochondrial glucose oxidation in BAT.

Activated glucose metabolism in BAT can also support fatty acids and triglyceride (TG) synthesis (Brito *et al*, 1999; Moura *et al*, 2005; Townsend & Tseng, 2014). Cold exposure significantly induced the m+3 enrichment of glyceraldehyde 3-phosphate (G3P), although the G3P level was reduced to 30% of that of mice housed at 30°C (Fig EV1C). These data suggested that, in addition to fueling the TCA cycle, glucose can also provide G3P for TG synthesis. Cold exposure also significantly induced the enrichment of palmitate, but the levels of enrichments were less than 0.5% in all isotopologues of palmitate, which were much lower than the enrichment of TCA cycle intermediates (Fig EV1D). These data suggested the activity of *de novo* fatty acids synthesis was very low in our experimental setting. Together with the high enrichment of TCA cycle intermediates, our data suggest chronic cold exposure increases glucose-dependent oxidation and glyceroneogenesis in BAT.

### Chronic cold exposure also induced oxidative metabolism in the sWAT, beige adipose tissue

Our previous studies showed that, in addition to activate BAT, chronic cold exposure also induced the "browning/beiging" of sWAT (Wang *et al*, 2013, 2015). We confirmed that, after housed at 6°C for 10 days, mice showed massive browning morphology in sWAT with positive UCP1 staining (Fig 2A). We then analyzed the cellular glucose metabolism in the sWAT of male mice. Different from BAT, the glucose enrichment in sWAT was not altered in male mice housed at 6°C (Fig 2B). Chronic cold exposure increased the m+3 enrichment of 3PG, but it did not alter the m+3 enrichment of

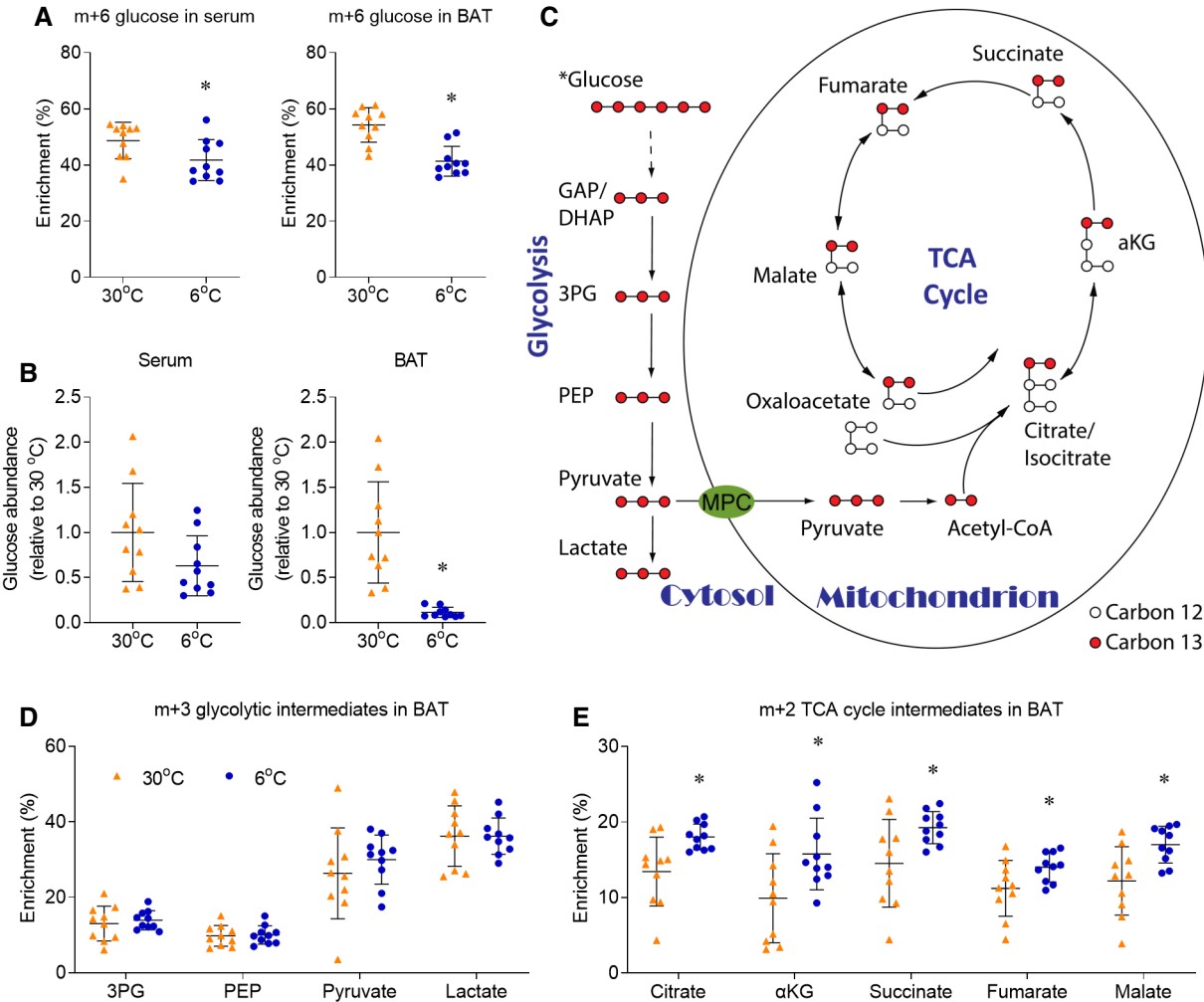

**Figure 1. Chronic cold exposure induces oxidative metabolism in BAT.**

Male mice, housed at 30°C or 6°C for 10 days, were intraperitoneally (IP) administered with [U-$^{13}$C]glucose (2 g/kg). 15 minutes after injection, BAT was harvested for metabolic enrichment assay.

A m+6 glucose enrichment in serum and BAT.

B Relative glucose abundance in serum and BAT.

C Illustration of the intracellular carbon transition in metabolic tissues after [U-$^{13}$C]glucose administration.

D, E Metabolic $^{13}$C enrichments in BAT of male mice are shown as m+3 glycolytic intermediates (D), m+2 TCA cycle intermediates (E).

Data information: $n = 10$, data are represented as the mean ± SD. Statistical analysis was performed using two-tailed Student's *t*-test, *$P < 0.05$. 3PG, 3-Phosphoglycerate; PEP, phosphoenolpyruvate; αKG, α-ketoglutarate.

Source data are available online for this figure.

other glycolytic intermediates (PEP, pyruvate and lactate) in the sWAT of male mice (Fig 2C). In comparison with largely unchanged glycolysis, chronic cold exposure significantly increased the enrichment of TCA cycle intermediates in the sWAT, beige adipose tissue (Fig 2D and Appendix Fig S2A).

Unlike sWAT, visceral WAT (vWAT) does not undergo dramatic browning upon cold exposure (Wu *et al*, 2012). [U-$^{13}$C]glucose tracing showed that chronic cold exposure repressed the enrichment of glycolytic intermediates in the gonadal WAT (gWAT, a typical vWAT) of male mice (Fig 2E and F). In comparison, chronic cold exposure did not change the m+2 enrichment of TCA cycle intermediates in gWAT (Fig 2G and Appendix Fig S2B), which was different

from BAT and sWAT. These data demonstrate that chronic cold exposure activates mitochondrial oxidative metabolism only in the white adipose tissue that had undergone browning/beiging and is capable for thermogenesis.

In addition to adipose tissue, muscle and liver also consume and metabolize glucose under physiological conditions. Chronic cold exposure significantly repressed glucose enrichment in the liver (Fig EV2A), which resulted in the reduced enrichment of downstream glycolytic and TCA cycle intermediates (Fig EV2B). After normalizing to the glucose enrichment in the liver of each mouse, the relative enrichments of the downstream metabolites were similar between 6°C and 30°C housed mice (Fig EV2C). If the similar

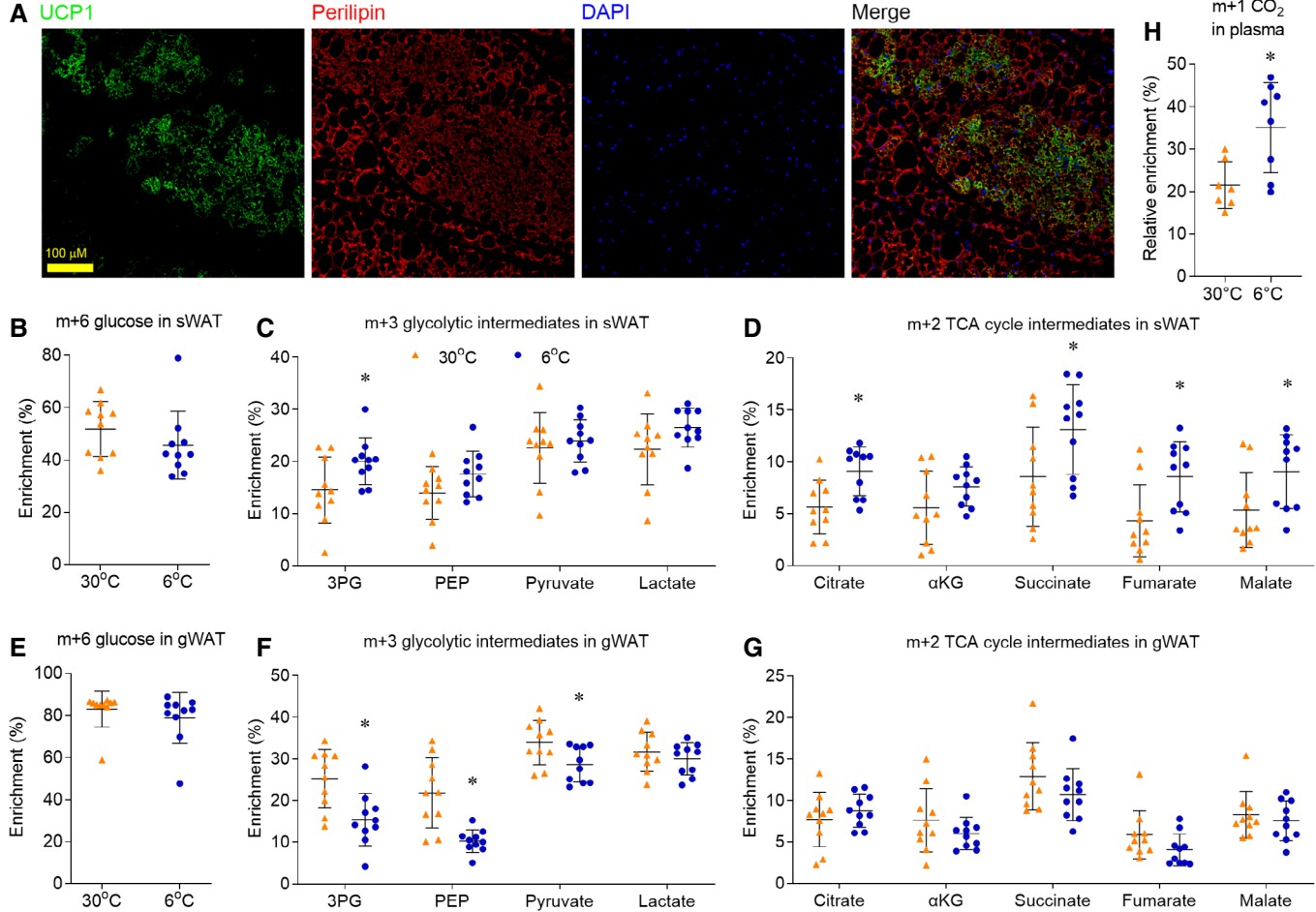

**Figure 2. Chronic cold exposure induces oxidative metabolism in sWAT.**

A    Representative immunofluorescence staining shows UCP1 (green), perilipin (red), and DAPI (blue) in sWAT of mice housed at 6°C.

B–G    Male mice, housed at 30°C or 6°C for 10 days, were administered with [U-$^{13}$C]glucose (2 g/kg, IP). 15 min after injection, sWAT and gWAT were harvested for metabolic enrichment assay. Metabolic $^{13}$C enrichments in sWAT of male mice are shown as m+6 glucose (B), m+3 glycolytic intermediates (C), and m+2 TCA cycle intermediates (D). Metabolic $^{13}$C enrichments in gWAT of male mice are shown as m+6 glucose (E), m+3 glycolysis intermediates (F), and m+2 TCA cycle intermediates (G).

H    Plasma $^{13}CO_2$ enrichment, after normalizing to the [U-$^{13}$C]glucose tracer enrichment.

Data information: $n = 10$ mice per group for (B-G), $n = 7-8$ mice per group for (H), data are represented as the mean ± SD. Statistical analysis was performed using two-tailed Student's *t*-test, *$P < 0.05$.

Source data are available online for this figure.

relative enrichments were calculated in BAT, the relatively low glucose enrichment would make the activation of glucose oxidation more dramatic in BAT.

At rest, muscle consumes a large amount of glucose in mammals. In mice housed at 30°C, glucose enrichment was higher in skeletal muscle (67%) than BAT (54%) (Fig EV2D), but the enrichment of glycolytic and TCA cycle intermediates was all relatively lower in skeletal muscle (Fig EV2E). These comparisons suggest that, under thermoneutrality, the basal oxidative metabolism in BAT had higher flux than that in the skeletal muscle. More importantly, chronic cold exposure only activated oxidative metabolism in BAT, but not in the muscle or liver. Altogether, these results indicate that, during chronic cold exposure, glucose oxidation is activated in BAT and beige adipose tissue, but not in WAT, muscle or liver.

## Chronic cold exposure induced oxidative metabolism in the BAT and sWAT of female mice

Although earlier BAT studies only included male subjects (Cypess *et al*, 2009; Lichtenbelt *et al*, 2009; Virtanen *et al*, 2009), it is important to know whether there is a gender difference in glucose metabolism upon chronic cold exposure. Similar to the experiment in male mice, the [U-$^{13}$C]glucose tracer was IP injected into female mice individually housed at 6 or 30°C for 10 days (Fig EV3 and Appendix Fig S3). Although cold exposure did not reduce the glucose enrichment in BAT of female mice housed at 6°C, the intracellular glucose metabolism in the BAT of female mice showed similar cold-induced changes as the BAT of male mice (Fig EV3A and B). Similar to the induction in male mice, chronic cold exposure also

increased the enrichment of m+3 G3P in the BAT of female mice (Fig EV3C).

Different from the increased m+3 enrichment of 3PG in the sWAT of male mice, chronic cold exposure decreased m+3 enrichment of 3PG, PEP, and pyruvate in female mice, which was resulted from the decreased glucose enrichment in sWAT of female mice housed at 6°C (Fig EV3D). Although chronic cold exposure altered the upper glycolytic pathways in different directions in female and male mice, it did not change the m+3 enrichment of lactate in the sWAT of both female and male mice (Fig EV3D). This suggests that, similar to BAT, cold exposure did not lead to enhanced glycolytic flux in sWAT. However, as in BAT, chronic cold exposure significantly increased the m+2 enrichment of TCA intermediates and m+3 enrichment of G3P in the sWAT of both male and female mice (Fig EV3E and F).

Similar to the gWAT of male mice, chronic cold exposure repressed the enrichment of glycolytic intermediates in the gWAT of female mice (Fig EV3G), without altering the m+2 enrichment of TCA cycle intermediates in the gWAT (Fig EV3H). Chronic cold exposure also increased the m+3 enrichment of G3P in the gWAT of female mice, not male mice (Fig EV3I). Together, these data suggested chronic cold exposure had similar effects on glucose-dependent oxidative metabolism in the adipose tissues in male and female mice. More importantly, we found the enrichment of $CO_2$, the final product of glucose oxidation, was significantly higher in the plasma of cold-exposed mice (Fig 2H).

### Differentiated primary brown adipocytes were more oxidative and less glycolytic

To better understand the cell-autonomous glucose metabolism of brown adipocytes, we performed [U-$^{13}$C]glucose tracing in the differentiated primary brown adipocytes cultured *in vitro*. Freshly isolated primary BAT stromal vascular fractions (SVF) were used for *in vitro* brown adipocyte differentiation. Oil Red O staining confirmed almost 100% differentiation at day 6, and the induction of brown adipocyte markers was also confirmed by RT–PCR (Fig 3A). Compared with the undifferentiated SVF, brown adipocytes had lower m+3 enrichment of 3PG, PEP, pyruvate, and lactate, indicating lower glycolytic activity (Fig 3B). The lower m+0 fractions of TCA cycle intermediates showed that glucose contribution to mitochondrial oxidative metabolism was more active in the differentiated brown adipocyte (Fig 3C). The increased m+2 enrichment of aKG, succinate, fumarate, and malate indicated the higher glucose oxidation in the differentiated brown adipocytes, and the increased m+3 enrichment of citrate, aKG, fumarate, and malate indicated higher pyruvate carboxylase (PC) activity in the differentiated brown adipocytes (Fig 3C). Induced PC expression has been reported in the differentiated 3T3-L1 adipocytes (Freytag & Utter, 1980), and the direct product of anaplerotic PC reaction is oxaloacetate, which can be incorporated with acetyl-CoA generated from fatty acid beta-oxidation. These data suggest that, comparing to the undifferentiated SVFs, brown adipocytes have higher glucose-dependent anaplerosis and glucose oxidation.

### Metabolic flux analysis (MFA) revealed that β3-AR agonist activated glucose oxidation

At day 6, *in vitro* differentiated brown adipocytes were stimulated with the β3-AR agonist (CL316,243), which mimics the *in vivo* cold stimulation (Held *et al*, 2018). β3-AR agonist treatment increased the expression of thermogenic genes (Fig 4A). β3-AR agonist treatment also significantly enhanced the enrichment of glycolytic intermediates (pyruvate and lactate, Fig 4B), and TCA cycle intermediates (Figs 4C and EV4). To better understand the metabolic effect of β3-AR agonist on brown adipocytes, we next used metabolic flux analysis (MFA) to integrate and model the enrichment data (Appendix Table S1). A set of conventional reactions and compartmentation were used to constrain the modeling (Jiang *et al*, 2016), and the metabolic fluxes were calculated from extracellular flux rates of glucose and lactate, and $^{13}$C distributions in several metabolites from the [U-$^{13}$C]glucose tracer. The MFA model showed that the fluxes of TCA cycle reactions were induced over two-fold by β3-AR agonist treatment, while the MPC and PDH were the most induced reactions (Fig 4D). The MFA model also indicated that the anaplerotic PC activity and fatty acid β oxidation were induced about two-fold (Appendix Table S1). Altogether, the β3-AR agonist treatment activated the oxidation of both glucose and fatty acids in the fully differentiated brown adipocytes, and the induction of glucose oxidation was higher than the induction of fatty acids oxidation. This effect of β3-AR agonist on the primary brown adipocytes was consistent with a recent study, which used similar MFA modeling to show that the β3-AR agonist could stimulate PDH activity in the differentiated T37i brown adipocyte cell line (Held *et al*, 2018). Our [U-$^{13}$C]glucose tracing experiment and mathematical modeling showed that the β3-AR agonist activated both cytosolic glycolysis and mitochondrial oxidation in the differentiated primary brown adipocytes *in vitro*.

### MPC inhibition blocked the oxidative metabolism in brown adipocytes and BAT

In both *in vivo* and *in vitro* models, our results showed that glucose oxidation was enhanced in brown adipocytes and BAT upon activation of thermogenesis. For glucose to be oxidized, it is first converted to pyruvate in the cytosol through glycolysis, and pyruvate can be further oxidized through the TCA cycle in the mitochondria. Cytosolic pyruvate enters mitochondria through the MPC, which is a heterodimeric complex formed by two subunits MPC1 and MPC2. Both MPC1 and MPC2 are highly expressed in tissues, including BAT, with greater mitochondrial abundance (Vigueira *et al*, 2014). The expression level of both MPC1 and MPC2 was significantly induced in BAT of the mice upon cold exposure, and in the brown adipocytes upon differentiation (Fig EV5A and B).

To further examine the role of MPC in brown adipocyte glucose oxidation, we used UK5099 and α-cyano-4-hydroxycinnamic acid (CHC) to block the mitochondrial pyruvate uptake (Schell *et al*, 2014; Yang *et al*, 2014). Both UK5099 and CHC treatment significantly repressed the oxygen consumption rate (OCR) in the β3-AR agonist-stimulated brown adipocytes (Figs 5A and EV5C). MPC inhibition also blocked β3-AR agonist-stimulated m+2 enrichment of TCA cycle intermediates, without altering the m+3 enrichment of glycolytic intermediates in brown adipocytes (Fig 5B). These results indicate that β3-AR agonist-stimulated mitochondrial oxidative metabolism requires MPC activity. Importantly, *in vivo* administration of CHC blocked the oxidative and glycolytic glucose metabolism in BAT of cold-exposed mice (Fig 5C). In addition to pyruvate oxidation, MPC inhibition also reduced the pyruvate carboxylation, glucose-dependent anaplerosis, as shown by the decreased m+3 enrichment of TCA cycle intermediate (Fig 5B and C). Similar to BAT, *in vivo* administration of CHC also

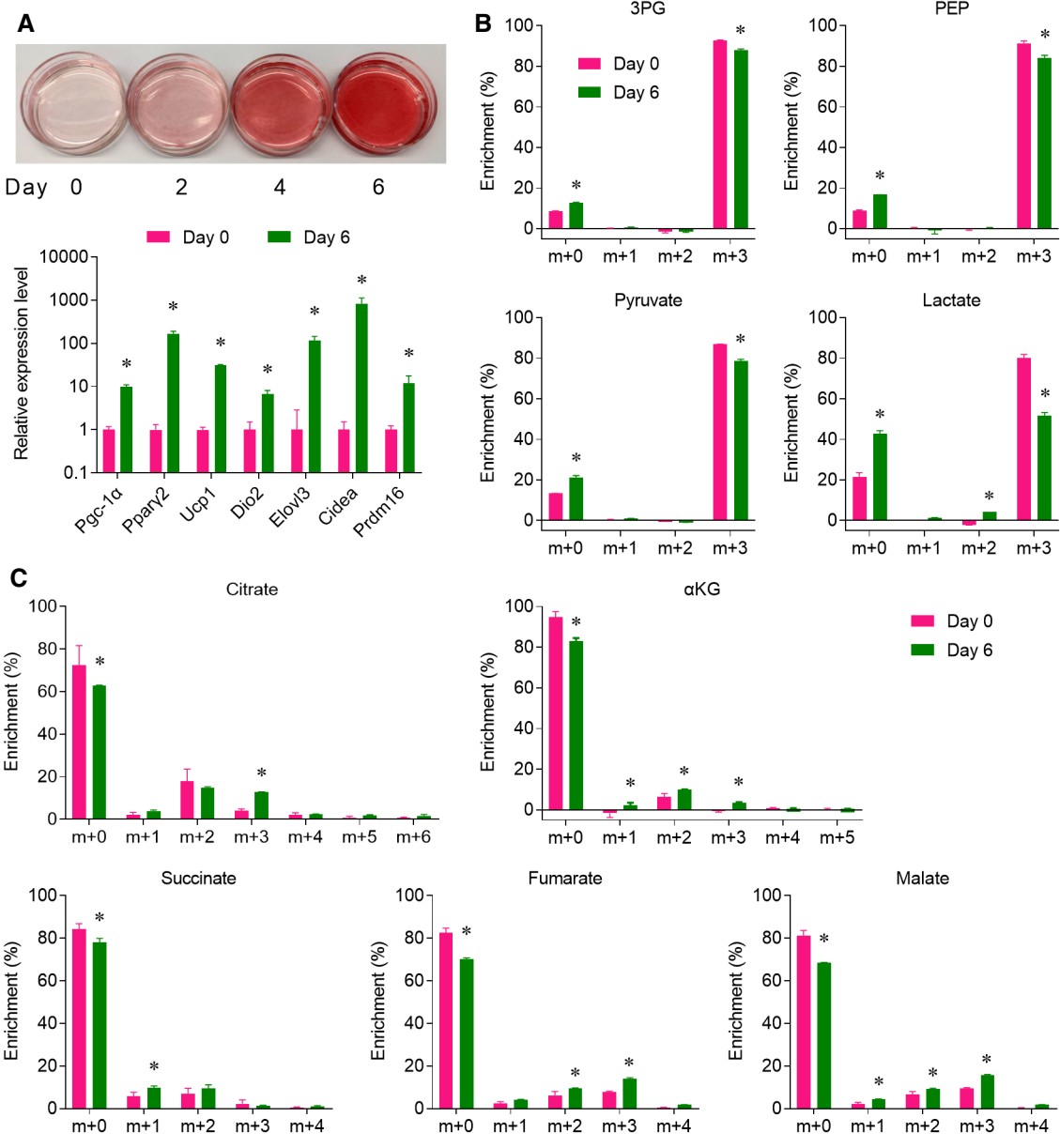

**Figure 3. Differentiated primary brown adipocytes are more oxidative and less glycolytic.**

Mouse brown adipocytes were differentiated from the freshly isolated cells in the stromal vascular fractions (SVF) of the interscapular BAT.

A Upper: Oil red staining of the cells during differentiation. Lower: Relative mRNA levels of brown adipocyte markers were measured by qPCR.

B, C (B) Glycolytic intermediates and (C) TCA cycle intermediates were analyzed by GC/MS, after cultured with medium containing 10 mM [U-$^{13}$C]glucose for 2 h.

Data information: $n = 3$ biological repeats, data are represented as the mean ± SD. Statistical analysis was performed using two-tailed Student's $t$-test (A), two-way ANOVA followed by Tukey's multiple comparisons test (B and C), *$P < 0.05$.

Source data are available online for this figure.

repressed the glucose metabolism in sWAT and gWAT of the cold-exposed mice (Fig EV5D and E). Most importantly, CHC administration impaired the body temperature maintenance of the mice upon cold exposure (Fig 5D). Notably, a recent study showed that mice with BAT-selective deletion of MPC1 also had significantly lower core body temperatures than their littermate controls (Panic *et al*, 2020), and the phenotype of this genetic model was consistent with the phenotype of our chemical MPC inhibition. Overall, these results

reveal that MPC-mediated glucose metabolism is a critical energy source for BAT and beige adipose tissue.

# Discussion

Similar to the regular glucose, [U-$^{13}$C]glucose is fully metaboliz-able. In comparison, the intracellular $^{18}$F-FDG can only be

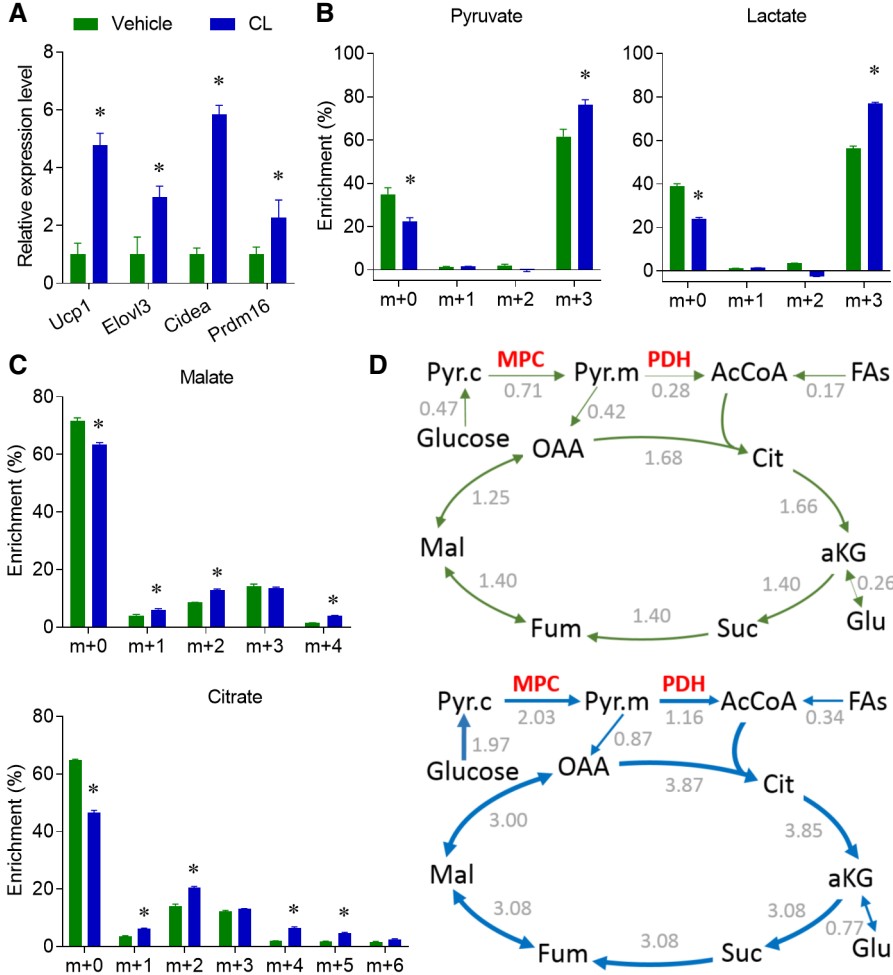

**Figure 4. β3-AR agonist activates glucose oxidation in differentiated primary brown adipocytes.**

A Fully differentiated brown adipocytes were treated with β3-AR agonist CL316,243 (10 μM). Relative mRNA levels of thermogenic markers were measured by qPCR.

B, C (B) Pyruvate and lactate enrichment, (C) Citrate and malate enrichment were analyzed by GC/MS, after cultured with medium containing 10 mM [U-$^{13}$C]glucose and 10 μM CL316,243 for 2 h.

D Results of metabolic flux analysis (MFA) of primary brown adipocytes treated with or without CL316,243. Arrows are weighted according to absolute flux magnitudes. See Appendix Table S1 for definitions of quantitative flux values, and the units of the flux values are nmol/h/μg protein. OAA, oxaloacetate.

Data information: *n* = 3 biological repeats, data are represented as the mean ± SD. Statistical analysis was performed using two-tailed Student's *t*-test (A), two-way ANOVA followed by Tukey's multiple comparisons test (B and C), *$P < 0.05$.

Source data are available online for this figure.

phosphorylated into $^{18}$F-FDG-6-phosphate. Because of the intracellular accumulation of $^{18}$F-FDG-6-phosphate, $^{18}$F-FDG-PET imaging is a great tool to visualize tissue with high glucose uptake activity. PET imaging with $^{11}$C-acetate is used to show active oxidative metabolism in BAT, as $^{11}$C-acetate-derived acetyl-CoA can be oxidized through mitochondrial TCA cycle. Although $^{18}$F-FDG-PET and $^{11}$C-acetate-PET indirectly indicated the complete glucose oxidation in BAT *in vivo*, $CO_2$ release from $^{14}$C-glucose assay showed the oxidation of glucose in the cultured brown adipocyte cell lines *in vitro* (Shackney & Joel, 1966; Irshad *et al*, 2017). It was not fully clear whether BAT could completely oxidize glucose *in vivo*. For the first time, we used the *in vivo* [U-$^{13}$C]glucose tracing to directly show the enhanced glucose oxidation in the BAT and beige adipose tissue of mice upon chronic cold

exposure, which supports a role for glucose oxidation in brown fat thermogenesis..

As [U-$^{13}$C]glucose tracer is fully metabolizable, *in vitro* [U-$^{13}$C] glucose tracing has been routinely used to study cellular metabolism in cultured cancer cell lines. More recently, *in vivo* [U-$^{13}$C]glucose tracing has been developed to study glucose metabolism in the tumor of patients with lung cancer (Hensley *et al*, 2016; Faubert *et al*, 2017). Although *in vivo* [U-$^{13}$C]glucose tracing has also been used to study glucose metabolism in adipose tissue (Nagao *et al*, 2017), the m+2 enrichment of TCA cycle intermediates was relative low in BAT (Mills *et al*, 2018). Here, we observed much higher enrichment of the TCA cycle intermediates after optimizing the experimental conditions. Our study is the first to show that chronic cold exposure activated glucose oxidation in both BAT and beige

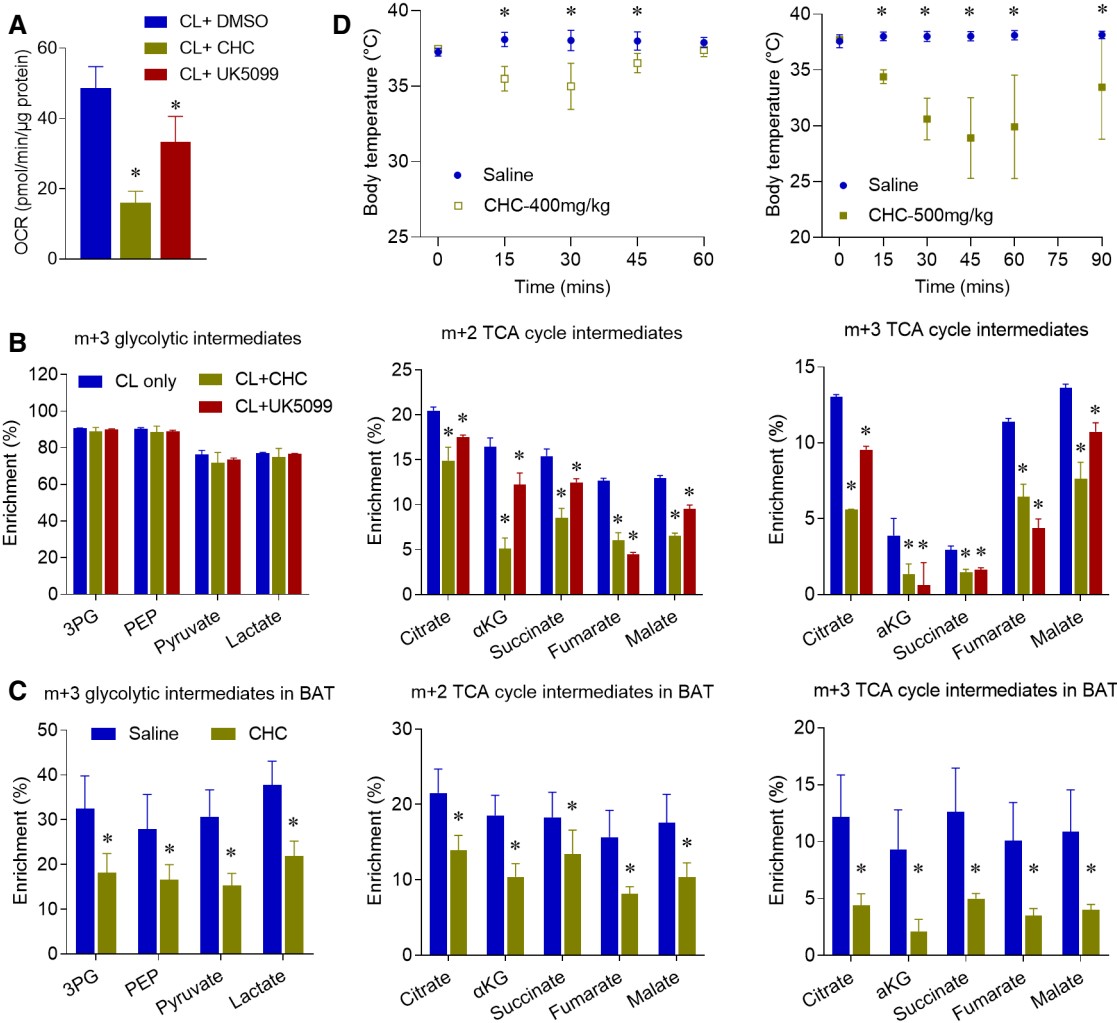

**Figure 5. MPC inhibition blocks oxidative metabolism in brown adipocytes and BAT.**

A   Oxygen consumption rate (OCR) of mouse brown adipocytes treated with MPC inhibitor CHC (2 mM) or UK5099 (2 μM), n = 6–7 biological repeats.

B   Differentiated mouse brown adipocytes were pre-treated with β3-AR agonist CL316,243 for 4 h, and then cultured with medium containing 10 mM [U-$^{13}$C]glucose for 2 h with CHC or UK5099. Metabolic $^{13}$C enrichments in brown adipocytes were shown as m+3 glycolytic intermediates, m+2, and m+3 TCA cycle intermediates, n = 3 biological replicates.

C   After housed at 6°C for 10 days, mice were IP injected with PBS or CHC (500 mg/kg). 30 min after CHC treatment, mice were IP administered with [U-$^{13}$C]glucose (2 g/kg). Metabolic $^{13}$C enrichments in BAT of male mice are shown as m+3 glycolytic intermediates, and m+2 and m+3 TCA cycle intermediates, n = 7.

D   Body temperature of the CHC (left: 400 mg/kg, n = 5) and (right: 500 mg/kg, n = 7) treated mice.

Data information: Data are represented as the mean ± SD. Statistical analysis was performed using one-way ANOVA followed by Tukey's multiple comparisons test (A and B), two-tailed Student's *t*-test (C), and two-way ANOVA followed by Sidak's multiple comparisons test (D), *$P < 0.05$.

Source data are available online for this figure.

adipose tissue. As over 40% of the TCA cycle intermediates in BAT was labeled upon a bolus IP injection of [U-$^{13}$C]glucose tracer (Appendix Fig S1), our results suggested that glucose served as an important energy source for mitochondrial oxidation in the BAT of mice upon chronic cold exposure.

In this study, we compared the effect of chronic cold exposure on glucose metabolism in multiple metabolic tissues. We found that chronic cold exposure induced glucose oxidation in both BAT and beige adipose tissue (sWAT), but not in gWAT, muscle or liver. The enrichment of glycolytic intermediates was all lower in gWAT and liver of mice upon chronic cold exposure, suggested that chronic

cold exposure might direct glucose toward the BAT and beige adipose tissue to support their energy need. We also found that the effect of chronic cold exposure on glucose-dependent glycolytic and oxidative metabolism was gender independent, as the alterations in all adipose tissues were similar between male and female mice.

The enhanced glucose oxidation in BAT upon cold exposure is fully mimicked by β3-AR agonist stimulation in the differentiated primary brown adipocytes *in vitro*. Our MFA modeling revealed the activated mitochondrial oxidation in β3-AR agonist-stimulated primary brown adipocytes, which was consistent with a recently non-stationary modeling of mitochondrial metabolism in the β3-AR-

stimulated T37i brown adipocyte cell line (Held *et al*, 2018). In that report, Held *et al* used multiple time points to stimulate a non-stationary MFA, and they found that PDH activity was significantly increased upon β3-AR activation (Held *et al*, 2018). As they primarily focused on the mitochondrial metabolism of β3-AR agonist-stimulated T37i cell line, our modeling of both cytosolic glycolysis and mitochondrial oxidation of β3-AR agonist-stimulated primary brown adipocytes provided additional knowledge about the glucose metabolism in the activated brown adipocytes.

Our MFA modeling revealed that MPC was one of the most induced fluxes in β3-AR-stimulated primary brown adipocytes. As illustrated in Fig 1C, MPC connected the cytosolic glycolysis and mitochondrial oxidation of the intracellular glucose metabolism. The role of MPC in the activated brown adipocytes and BAT was experimental verified with the treatment of MPC inhibitors, UK5099 and CHC. Most importantly, *in vivo* CHC administration inhibited glucose oxidation and impaired the body temperature maintenance of the mice upon cold exposure. Unlike the complete loss of body temperature control in the mice with BAT-selective deletion of MPC1 (Panic *et al*, 2020), *in vivo* CHC administration only transiently reduced the body temperature of cold-exposed mice. One possibility is that bolus injection of CHC could only temporally block mitochondrial pyruvate uptake in BAT. Another possibility is that lack of MPC activity could be compensated by pyruvate-alanine cycling, which has been observed in liver-specific MPC mice (McCommis *et al*, 2015). Together, mitochondrial pyruvate uptake and oxidation serves an important energy source in the chronic cold exposure activated BAT and beige adipose tissue, which supports a role for glucose oxidation in brown fat thermogenesis.

It is worth noting that the single time point of enrichment measurement might not truly reflect the glucose kinetics under steady state conditions, especially for the TCA cycle intermediates. For the *in vitro* cultured brown adipocytes, it will be ideal to monitor the metabolic enrichment from multiple time points, with multiple additional tracers for palmitate, lactate and TG. Integrated MFA modeling with multiple tracers will be a better approach to reflect the intracellular metabolism of brown adipocytes. Different from *in vitro* culture, glucose tracer is hard to reach fully enrichment *in vivo*, because of the high level of pre-existing unlabeled glucose. Since all tissues, including BAT, use both endogenous unlabeled glucose and exogenous labeled glucose tracer, the enrichment of interested metabolites can be shown as the relative enrichment by normalizing to the tracer enrichment (Fig EV2C). As previous reported (Nagao *et al*, 2017), a bolus IP injection of 2 g/kg [U-$^{13}$C] glucose tracer might increase plasma glucose, which might alter the BAT glucose metabolism. A long time infusion of low level of [U-$^{13}$C]glucose tracer is a better approach to reach high glucose enrichment, without altering the circulating metabolites (Hensley *et al*, 2016). As the surgery for infusion approach might be too invasive for the cold-exposed mice, it will be interesting to assay the alteration of glucose metabolism in CL-activated BAT in the future studies.

In addition to glucose, fatty acids and TG are believed to act as the major fuels for BAT thermogenesis (Townsend & Tseng, 2014), and intracellular TG has been reviewed as the primary energy source upon acute cold exposure (Ma & Foster, 1986; Baba *et al*, 2010; Labbe *et al*, 2015; Blondin *et al*, 2017). One study showed that the total lipid content of BAT was significantly reduced after

cold exposure (Baba *et al*, 2010). Another study performed the glucose tracing assays in mice upon (3 h) acute cold exposure, and there were no changes in the enrichment of glycolytic and TCA cycle intermediates (Mills *et al*, 2018). It is also worth to note that mice defective in brown fat lipolysis are not cold-sensitive (Shin *et al*, 2017), and BAT had the flexibility in glucose and fata utilization (Wang *et al*, 2019). It is likely that, when intracellular TG is reduced after chronic 10 days cold exposure, glucose acts as another fuel for BAT.

Several studies showed that MPC inhibition can induce compensatory metabolic pathways (Taylor, 2017), and one study showed that the MPC1 +/− mice might employ fatty acid oxidation to meet their bioenergetic demands (Zou *et al*, 2018). Our *in vivo* [U-$^{13}$C] glucose tracing study showed that over 50% of the TCA cycle intermediates in BAT were the unlabeled m+0 isotopologues, which could be produced from other nutrients, including fatty acids. Enhanced fatty acids uptake in BAT upon cold exposure was observed by the $^{18}$FTHA-PET imaging in both rodent models and humans (Labbe *et al*, 2015; Caron *et al*, 2017; Din *et al*, 2018). It will be interesting to use a similar stable isotope tracing approach to quantify the contribution of fatty acids and intracellular TG to BAT thermogenesis. A better understanding of the glucose and fatty acids metabolism in BAT could potentially identify new therapeutic targets to treat obesity through enhancing thermogenesis.

# Materials and Methods

### Animals studies

All animal studies were approved by the Institutional Animal Care and Use Committees of city of hope. All animals in this study were male and female C57BL/6J background (9–12 weeks) and housed on a 12 h dark/light cycle in a temperature-controlled room (24°C) (Song *et al*, 2020).

### Mouse primary brown adipocytes preparation and differentiation

Mouse BAT SVF was obtained from the interscapular brown fat pad of mice at postnatal day 5–8 (Mills *et al*, 2018). Briefly, dissected brown adipose tissues were cut into small pieces and digested for 45 min at 37°C in digestion buffer (2× stock solution containing 0.123 M NaCl, 5 mM KCl, 1.3 mM CaCl$_2$, 5 mM glucose, 100 mM HEPES, and 4% BSA, add equal volume PBS and 1 mg/ml collagenase D (Roche Diagnostics Corporation, Indianapolis, IN, USA). Digested cell/tissue mixture was filtered through 100 μm cell strainer (431752, Corning, Corning, NY, USA) to remove undigested tissues and then centrifuged for 5 min at 600 *g* at 4°C. The cell pellet was resuspended in culture medium DMEM (10566016, Thermo Fisher Scientific, Waltham, MA, USA); 1% penicillin/streptomycin (SV30010, Hyclone, Pittsburgh, PA, USA), 0.1% gentamicin (1855724, Life technologies Waltham, MA, USA), 20 mM HEPES (H3537,Sigma, St. Louis, Missouri, USA), and 20% FBS (16H328, Sigma, St. Louis, Missouri, USA)) and then filtered through 40 μm cell strainer (431750, Corning, Corning, NY, USA) to remove clumps and large adipocytes. Centrifuged for 5 min at 600 *g* at 4°C, SVF pellet was then resuspended in culture medium and plated onto 60 mm dishes. Subculture cells when cells reach 80-90% confluence.

For mouse brown adipocyte differentiation, day 0 (2 days after reaching confluence) cells were switched to fresh induction media with dexamethasone (1 μM); insulin (0.12 μg/ml); isobutylmethylxanthine (0.5 mM); 3,3′,5-Triiodo-L-thyronine (1 nM); indomethacin (0.125 mM). After 2 days, the medium was changed to maintenance medium containing insulin (0.12 μg/ml); 3,3′,5-Triiodo-L-thyronine (1 nM); and changed every 2 days until harvest. Cells were fully differentiated by day 6.

### Oil red O staining

Cells were washed twice with PBS, fixed with 10% formalin at RT at least for 1 h, washed twice with ddH$_2$O and once with 60% isopropanol for 5 min and then dried at RT. Cells were stained with Oil Red O (O-0625, Sigma, St. Louis, Missouri, USA) for 10 min. Excess stain was removed immediately with ddH$_2$O, and images were acquired under the microscope.

### Respirometry detection of primary brown adipocytes

The cellular OCR was detected by a Seahorse XF24 Extracellular Flux Analyzer as described previously with minor changes (Mills *et al*, 2018). 10,000 cells for each well were plated in an XF24V7 cell culture microplate (cat# 100777-004) and started differentiation as described above after 12 h. Prior analysis, the medium was changed to XF Base Medium (cat# 102353-100) containing 25 mM glucose and 3 mM glutamine (pH 7.40). 100 nM CL316,243 was added, and respiration was detected after basal respiration detection. Then, the OCR changes induced by MPC inhibition were detected after 2 mM CHC or 2 μM UK5099 addition.

### *In vitro* [U-$^{13}$C]glucose tracing

For mouse primary brown adipocytes, glucose tracing experiments were conducted on Day 0 and Day 6. At different time points, the medium was removed completely and changed to tracing medium (D5030, Sigma, St. Louis, Missouri, USA) containing 1% HEPES, 1% pen/strep, 10 mM [U-$^{13}$C]glucose, and 3 mM glutamine for 2 h (Jiang *et al*, 2017). After fully differentiation (Day 6), mouse brown adipocytes were treated with PBS or 10 μM β3-AR agonist CL316,243 (C5976, Sigma, St. Louis, Missouri, USA) for 4 h and then changed to tracing medium containing DMSO or 10 μM CL316243 co-treating with or without MPC inhibitor 2 mM α-Cyano-4-hydroxycinnamic acid (CHC, 476870, Sigma, St. Louis, Missouri, USA) and 2 μM Uk5099 (PZ0160, Sigma, St. Louis, Missouri, USA) for 2 h. After incubated with tracing medium for 2 h, cells in 35 mm dishes were washed once with cold NaCl (0.9% w/v), harvested with 800 μl cold 50% methanol into 1.5 ml tubes, and then frozen in liquid nitrogen. Then, the samples were frozen/thawed for 3 times and centrifuged at 4°C, 18,000 *g* for 15 min. The supernatant was transferred and dried down for further GC/MS sample preparation. Tissues pellet was dissolved with 0.1 M NaOH for BCA protein assay.

### *In vivo* [U-$^{13}$C]glucose tracing

After individually housed at 6 or 30°C for 10 days, mice were administered with [U-$^{13}$C]glucose (2 g/kg) intraperitoneal (IP) injection (Fan *et al*, 2011; Nagao *et al*, 2017). 15 minutes after injection, mice were sacrificed, and tissues were collected and snap frozen in liquid nitrogen. Tissues (15–30 mg for BAT, liver, and muscle; whole fat pad for subcutaneous WAT and gonadal WAT) were homogenized with tissue lyser in 80% methanol on ice. Then, the samples were frozen/thawed for 3 times and centrifuged at 4°C, 18,000 *g* for 15 min. The supernatant was transferred and dried down for further GC/MS sample preparation. Tissues pellet was dissolved with 500 μl 0.1 M NaOH for BCA protein assay.

### Mass spectrometry analysis

Dried metabolites were derivatized for 2 h at 42°C in 50 μl of methoxyamine hydrochloride 10 mg/ml (Sigma) and 100 μl N-tert-Butyldimethylsilyl-N-methyltrifluoroacetamide (Sigma) for 90 min at 72°C (Dai *et al*, 2020). Metabolites were analyzed using an Agilent 7890B gas chromatograph (Agilent, CA, USA) networked to an Agilent 5977B mass selective detector. Retention times and mass fragmentation signatures of all metabolites were validated using pure standards. To determine the relative metabolite abundance across samples, the area of the total ion current peak for the metabolite of interest was normalized for protein content. The mass isotopomer distribution analysis measured the fraction of each metabolite pool that contained every possible number of 13C atoms: a metabolite could contain 0, 1, 2, ...n $^{13}$C atoms, where n = the number of carbons in the metabolite. For each metabolite, an informative fragment ion containing all carbons in the parent molecule was analyzed by MATLAB software (MathWorks, CA, USA). The abundance of all mass isotopomers was integrated from m+0 to m+n, where m = the mass of the fragment ion without any $^{13}$C. The abundance of each mass isotopomer was then corrected mathematically to account for natural abundance isotopes and finally converted into a percentage of the total pool.

### Measurement of plasma $^{13}$CO$_2$ enrichment

The enrichment, Atom % Excess (APE) in $^{13}$CO$_2$ was determined by the Metabolomics Core Facility of The Children's Hospital of Philadelphia Research Institute as in (Nissim *et al*, 2006). Briefly, 10–20 μl of plasma was added to a sealed tube free of CO$_2$, and then, flashed with helium for 5 min. Then, 100 μl of 20% phosphoric acid was added, and the sample was vortexed to liberate $^{13}$CO$_2$. The latter was removed with a sealed syringe and transferred to auto-sampler tubes for analysis. Isotopic enrichment in $^{13}$CO$_2$ was determined by Isotope Ratio Mass Spectrometry (Thermo Fisher Delta V). using the *m/z* 45/44 ratio. Quality control of each sample was maintained by measuring the 45/44 ratio of reference gas (pure CO$_2$) as well as standard gas (0.3% CO$_2$ in helium).

### Metabolic flux analysis

Extracellular flux rates (glucose consumption, lactate secretion) and mass isotopologues distributions (MID) of pyruvate, lactate, citrate, αKG, succinate, fumarate, malate, glutamate, glutamine, and palmitate were combined to calculate steady state metabolic flux by using INCA software package (Young, 2014). Reaction networks describing the flux of central carbon metabolism were developed with assumptions described before (Jiang *et al*, 2016), and additional

assumptions were described in the legend of Appendix Table S1. Data used for MFA were reported in the Appendix Table S2. To ensure that a global minimum of fluxes was identified, flux estimations were initiated from random values and repeated a minimum of 50 times. A chi-square test was applied to test goodness-of-fit, and accurate 95% confidence intervals were calculated by assessing the sensitivity of the sum of squared residuals to flux parameter variations. Values, the lower and upper bounds of 95% confidence intervals of all fluxes, were shown in Appendix Table S1.

### Body temperature

After individually housed at 6°C for 10 days, mice were administered with CHC (400 mg/kg or 500 mg/kg) through IP injection (Yang *et al*, 2014). Rectal temperature was measured every 15 min after CHC injection.

### Real-time PCR

Total RNA was extracted from tissues using TRIzol (15596-026, Thermo Fisher Scientific, Waltham, MA, USA) and reversed to cDNA with iScript Reverse Transcription Kit (1708840, Bio-Rad, Berkeley, CA, USA). Real-time PCR was processed on cDNA using SYBR Green (4472908, Thermo Fisher Scientific, Waltham, MA, USA). Relative mRNA levels were calculated with Rplpo as the internal control. The primers are listed in the Appendix Table S3.

### Immunofluorescence staining

Formalin-fixed, paraffin-embedded sections from adipose tissue (*n* = 2–3 male mice) were blocked in PBST with 5% BSA (Wang *et al*, 2015). Primary antibody used was perilipin (1:500 dilution) (NB100-60554, NOVUS) and UCP1 (1:250 dilution) (ab10983, Abcam, Cambridge, England); secondary antibodies (1:200 dilution) used were Alexa Fluor 594 Donkey anti-Goat IgG (H+L) and Alexa Fluor 488 Donkey anti-Rabbit IgG (H+L) (Invitrogen, Carlsbad, CA, USA). Slides were counterstained with DAPI. Images were acquired using AxioObserver Epifluorescence Microscope (Zeiss, Jena, Germany).

## Data availability

All data needed to evaluate the conclusions in the paper are present in the paper and/or the Appendix Materials. No data were deposited in a public database. Additional data related to this paper may be requested from the authors.

**Expanded View** for this article is available online.

## Acknowledgements
We thank Y. Daikhin, O. Horyn, and Ilanna Nissim for performing the measurements of $^{13}CO_2$ in the Metabolomics Core Facility, The Children's Hospital of Philadelphia (http://www.research.chop.edu/cores/metabolomic/). We thank the City of Hope Animal Resource Center for mouse study assistance. We thank Leonard Medrano at City of Hope for his help with OCR measurements. We are grateful to Ralph DeBerardinis at UTSW for discussions and comments. L.J. was supported by City of Hope Medical Center Start-up and P30CA033572. Q.A.W. was supported by US National Institutes of Health grants K01DK107788, R03HD095414, R56AG063854, R01AG063854, and R01HD096152. Q.A.W. was also supported by American Diabetes Association Junior Faculty Development Award 1-19-JDF-023.

## Author contributions
QAW and LJ designed the experiments and wrote the manuscript. ZW, TN, and AS prepared primary brown adipocytes and handled the *in vivo* mice experiments. ZW and LJ performed the GC/MS analysis and prepared MFA modeling. JR contributed to the experiment discussion and edited manuscript.

## Conflict of interest
The authors declare that they have no conflict of interest.

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
