## [Review Process File · EMBO Reports]

Chronic cold exposure enhances glucose oxidation in brown adipose tissue

Zhichao Wang, Tinglu Ning, Anying Song, Jared Rutter, Qiong Wang and Lei Jiang

DOI: 10.15252/embr.202050085

Corresponding author(s): Lei Jiang (ljiang@coh.org) , Qiong Wang (qwang@coh.org)

Review Timeline:

Submission Date:	22nd Jan 20
Editorial Decision:	4th Mar 20
Revision Received:	17th Jul 20
Editorial Decision:	19th Aug 20
Revision Received:	2nd Sep 20
Accepted:	10th Sep 20

Editor: Deniz Senyilmaz Tiebe

Transaction Report:

Dear Dr. Jiang,

Thank you for submitting your manuscript for consideration by EMBO Reports. My apologies for the delay in getting back to you, it took longer than anticipated to receive the referee reports. Initially three referees agreed to review this manuscript. So far, we have received two referee reports that are copied below. Given that both referees are in fair agreement that you should be given a chance to revise the manuscript, I would like to ask you to begin revising your study along the lines suggested by the referees.

Please note that this is a preliminary decision made in the interest of time, and that it is subject to change should the third referee offer very strong and convincing reasons for this. As soon as/if we receive the final report on your manuscript, we will forward it to you as well.

As you can see, the referees find analysis on the fate of glucose in stimulated BAT of interest. However, they also raise a number of concerns that need to be addressed to consider publication here. In particular, the referees require

- further insight into the fate of glucose (at the level of CO₂ analysis and non-oxidative fate of glucose (ref #1 points 1 and 3, ref #2 point 1)
- stronger support into the BAT specific requirement of MPC mediated glucose import into the mitochondria for efficient thermogenesis (ref #1 point 2 and ref #2 point 2).

I find the reports informed and constructive, and believe that addressing the concerns raised will significantly strengthen the manuscript.

Given these constructive comments, we would like to invite you to revise your manuscript with the understanding that the referee concerns (as in their reports) must be fully addressed and their suggestions taken on board. Please address all referee concerns in a complete point-by-point response. Acceptance of the manuscript will depend on a positive outcome of a second round of review. It is EMBO reports policy to allow a single round of revision only and acceptance or rejection of the manuscript will therefore depend on the completeness of your responses included in the next, final version of the manuscript.

1. A data availability section providing access to data deposited in public databases is missing (where applicable).
2. Your manuscript contains statistics and error bars based on n=2 or on technical replicates. Please use scatter plots in these cases.

Supplementary/additional data: The Expanded View format, which will be displayed in the main

HTML of the paper in a collapsible format, has replaced the Supplementary information. You can submit up to 5 images as Expanded View. Please follow the nomenclature Figure EV1, Figure EV2 etc. The figure legend for these should be included in the main manuscript document file in a section called Expanded View Figure Legends after the main Figure Legends section. Additional Supplementary material should be supplied as a single pdf labeled Appendix. The Appendix includes a table of content on the first page with page numbers, all figures and their legends. Please follow the nomenclature Appendix Figure Sx throughout the text and also label the figures according to this nomenclature. For more details please refer to our guide to authors.

2) individual production quality figure files as .eps, .tif, .jpg (one file per figure).

3) a .docx formatted letter INCLUDING the reviewers' reports and your detailed point-by-point responses to their comments. As part of the EMBO Press transparent editorial process, the point-by-point response is part of the Review Process File (RPF), which will be published alongside your paper. For more details on our Transparent Editorial Process, please visit our website:

<https://www.embopress.org/page/journal/14693178/authorguide#transparentprocess>

4) a complete author checklist, which you can download from our author guidelines (). Please insert information in the checklist that is also reflected in the manuscript. The completed author checklist will also be part of the RPF.

5) Please note that all corresponding authors are required to supply an ORCID ID for their name upon submission of a revised manuscript (). Please find instructions on how to link your ORCID ID to your account in our manuscript tracking system in our Author guidelines ().

6) We replaced Supplementary Information with Expanded View (EV) Figures and Tables that are collapsible/expandable online. A maximum of 5 EV Figures can be typeset. EV Figures should be cited as 'Figure EV1, Figure EV2' etc... in the text and their respective legends should be included in the main text after the legends of regular figures.

7) We would also encourage you to include the source data for figure panels that show essential data.

Numerical data should be provided as individual .xls or .csv files (including a tab describing the data). For blots or microscopy, uncropped images should be submitted (using a zip archive if multiple images need to be supplied for one panel). Additional information on source data and instruction on how to label the files are available .

8) Our journal encourages inclusion of *data citations in the reference list* to directly cite datasets that were re-used and obtained from public databases. Data citations in the article text are distinct from normal bibliographical citations and should directly link to the database records from which the data can be accessed. In the main text, data citations are formatted as follows: "Data ref: Smith et al, 2001" or "Data ref: NCBI Sequence Read Archive PRJNA342805, 2017". In the Reference list, data citations must be labeled with "[DATASET]". A data reference must provide the database name, accession number/identifiers and a resolvable link to the landing page from which the data can be accessed at the end of the reference. Further instructions are available at .

9) Please make sure to include a Data Availability Section before submitting your revision - if it is not applicable, make a statement that no data were deposited in a public database. Primary datasets (and computer code, where appropriate) produced in this study need to be deposited in an appropriate public database (see).

The accession numbers and database should be listed in a formal "Data Availability " section (placed after Materials & Method) that follows the model below. Please note that the Data Availability Section is restricted to new primary data that are part of this study.

Data availability

10) Regarding data quantification, please ensure to specify the name of the statistical test used to generate error bars and P values, the number (n) of independent experiments underlying each data point (not replicate measures of one sample), and the test used to calculate p-values in each figure legend. Discussion of statistical methodology can be reported in the materials and methods section, but figure legends should contain a basic description of n, P and the test applied.

Please note that error bars and statistical comparisons may only be applied to data obtained from at least three independent biological replicates.

I look forward to seeing a revised version of your manuscript when it is ready. Please let me know if you have questions or comments regarding the revision.

Yours sincerely,

Deniz Senyilmaz Tiebe

Deniz Senyilmaz Tiebe, PhD
Editor
EMBO Reports

Referee #1:

Brown adipose tissue, which is specialized in heat generation to keep a constant body temperature, uptakes large amount of glucose when it is activated. It is well established that free fatty acids are physiological activator of UCP1 and the main fuel of brown adipocytes. The metabolic fate of glucose is unclear. Theoretically, glucose has different metabolic fates in activated brown adipocytes including glycolytic ATP production, anaplerotic reactions in mitochondria, de novo fatty acids synthesis and fatty acid reesterification. The manuscript by Ning et al., traced the metabolic fate of glucose in brown fat in vivo through isotope labeling. The authors found that 10-days cold exposure dramatically increased enrichment of two-carbon (m+2) labeled TCA cycle intermediates, while did not alter the enrichment of glycolytic intermediates in brown fat. By injection of inhibitors of mitochondrial pyruvate carrier (MPC) to the mice, the authors observed that m+2 labeled TCA cycle intermediates were decreased in brown fat and body temperature homeostasis was impaired. Consequently, the authors concluded that glucose is oxidized and required for thermogenesis in cold-activated brown fat. To further improve the manuscript, the following issues are needed to be addressed.

Major:

1. Based on the current results, glucose is metabolized rather than oxidized. One possibility is that glucose is metabolized for de novo fatty acids synthesis. To provide evidence that glucose is oxidized, CO₂ need to be collected and analyzed its carbon labeling.
2. Injection of MPC inhibitors into mice was a whole body effect. MPC inhibitors target every organ. Therefore, the body temperature phenotype could not be considered as a direct consequence of impaired BAT thermogenesis. Furthermore, the authors need to demonstrate whether the thermogenesis function of brown fat is affected by MPC inhibition by monitoring brown fat temperature or brown fat cell thermogenesis.
3. How about the labeling information of other metabolites such as G3P, fatty acids? Does the glucose only function to fuel TCA cycle in brown fat?
4. As fatty acids are believed to act as the main fuels for brown fat thermogenesis, it is therefore of interest to discuss why brown fat mitochondria metabolize glucose to fuel TCA cycle in parallel. The authors should discuss possible reasons in the discussion part.

Minor:

1. How enrichment is calculated? How about total abundance? These information need to be provided.

2. what's the N number for cell culture experiments? n=3 is 3 biological experiments or technical replicates?
3. Brown fat uptakes glucose upon acute activation. The authors explored the metabolic fate of glucose in 10-days cold-acclimated mice. How about the acute activation condition? Is the metabolic fate of glucose different from cold-acclimated mice?
4. In fig 6c, it seems that the body temperatures of MPC blocked mice were recovered gradually, which indicates compensatory adaptation. Could the authors please comment on this point?
- 5 Line 486, differentiated should be Differentiated.

Referee #3:

In this study, Ning et al. investigate the metabolic fate of circulating glucose in cold- or pharmacologically-stimulated brown adipocytes. The authors first gave a bolus i.p injection of [U13C]glucose in male and female mice individually housed at 6{degree sign}C (cold exposure) or 30{degree sign}C (thermoneutral) for 10 days and examined circulating and tissue enrichment of 13C-labelled glucose and the subsequent distribution of the label within glycolytic and TCA cycle intermediates. The authors demonstrate that although circulating glucose enrichment is the same in thermoneutral and cold exposed mice, BAT glucose enrichment of the cold exposed mice was less than 10% of the thermoneutral mice, which the authors posit may reflect the increased glucose oxidation rate in BAT in the cold, as suggested by the increased enrichment of TCA cycle intermediates. The enrichment of TCA cycle intermediates also increased in the subcutaneousWAT that appears to have 'browned' in cold exposed mice, but not in the gonadaWAT (a more classic WAT depot). The authors then used metabolomics flux analysis to examine the metabolic fate of glucose in differentiated brown adipocytes stimulated with a beta3-adrenergic receptor agonist (CL316,243) without and with the inhibition of the mitochondrial pyruvate carrier (MPC), which is highly induced in cold-exposed mice and plays an important role in ensuring the oxidation of pyruvate. Indeed, the inhibition of MPC suppressed the enrichment of TCA cycle intermediates. The authors conclude that glucose oxidation is indispensable to BAT thermogenesis in mice.

Critique:

This is a very elegantly performed study that is certainly very timely for the field, well written and good description of methods. It is nice to see experiments performed in male and female mice and data presented disaggregated. The metabolomics approaches used provide valuable insight into the oxidative fate of glucose in both unstimulated and stimulated BAT, 'browned/beiged' sWAT and gWAT. The oxidative fate of glucose in BAT is of particular interest and the results gained will prove invaluable not only for future studies applying similar methods, but also in the interpretation of results generated by FDG PET or in the development of other PET radiotracers interested in examining BAT oxidative metabolism. There are, however, several critical gaps that warrant further clarification, some of which can likely be addressed through additional experiments.

Major

1) The in vivo measures of glucose flux in BAT and its subsequent metabolic fate provides a valuable extension of the currently available data derived in vitro. However, there are two critical gaps that could assist in completing the story. First, the lack of whole-body CO₂ production, expired ¹³CO₂ and tissue ¹³CO₂ leaves a significant gap and therefore leads to several unverified assumptions within the study. If interested in determining the rate of oxidation of ¹³C-labeled glucose, you must at least determine the rate of ¹³CO₂ produced either at the whole-body and/or by the tissue. It is difficult to investigate the oxidation of a substrate without knowing the rate with which the final downstream product (CO₂) is produced. Several inferences could be made with the

measure of whole-body VCO₂ and ¹³CO₂. Both are also simple measures to make with any metabolic system. If you assume a large proportion of the increased thermogenesis after 10 d of cold exposure is due to BAT, then several inferences could be made about ¹³CO₂/totalCO₂ production and its relationship to BAT glucose oxidation. Fick-derived measures of tissue ¹³CO₂ would be helpful too, but more difficult to execute. The addition of indirect calorimetry, to quantify total VCO₂ and substrate utilisation, combined with measures of ¹³C carbon dioxide enrichment (¹³CO₂) would provide valuable context to the current data.

The second gap is in examining the non-oxidative fate of the glucose (de novo fatty acid synthesis, glyceroneogenesis). Several lines of evidence from in vitro experiments (PMID: 27110487), inferences made from cold-acclimation-induced changes in gene expression (PMID: 25681456) and in vivo studies (PMID: 15688247, 10198378) suggest that a large proportion of glucose taken up by BAT under stimulated conditions is used for de novo fatty acid synthesis and glyceroneogenesis, to support the high intracellular TG turnover. The reductionist perspective presented here omits this critical fate of glucose. This manuscript would benefit from examining at least the enrichment of glycerol-3-phosphate/glycerol and FA isotopologues in BAT to provide valuable context.

2) The conclusion that glucose oxidation is required for thermogenesis in the cold-activated brown adipose tissue is a significant overstatement of the study findings and not fully supported by the data provided. Glucose oxidation in BAT may indeed increase in the cold, but the evidence supporting that it is required for thermogenesis is lacking. The fact that T_{core} (Figure 6C) appears to recover over time following MPC inhibition, which is the only data available to support this premise, seems to refute that conclusion (shows the effect is transient). As total energy expenditure increases ~5-fold (again, could be measured by indirect calorimetry) or more at such an intense cold, does it not make sense that glucose-derived TCA cycle intermediates also increase? Quantitatively, this may not be all that significant as iTG is the predominant fuel source. Fig 4 suggests that the cold-induced increase in glucose oxidation likely accounts for ~10% of total carbon flux through TCA cycle (citrate m+0 went down ~10% and m+3 went up ~10%). What both the in vivo and in vitro data suggests is that a certain proportion of glucose taken up by the tissue is directed towards oxidation and that this flux increases in the cold or with CL-treatment. It could be worth considering Seahorse experiments to examine the effects of inhibiting MPC (and PDH) on OCR.

Minor

- 1) In Fig. 6, it would be useful to have the enrichment of the glycolytic metabolites (upstream of the inhibition).
- 2) Figures 2 and 3, would benefit from presenting tissue enrichments of ¹³C-glucose.
- 3) On Line 244 - the authors report that '...over 40% of the TCA cycle intermediates in BAT was labeled upon a bolus IP injection of [U-¹³C]glucose tracer, our results suggested that glucose served as an important energy source for mitochondrial oxidation in the BAT of mice upon chronic cold exposure.' It is unclear where this 40% is derived from. Figure 1E shows 20%?
- 4) Fig 6. seems to also suggest that there may be effects on anaplerotic pathways - has this been examined?
- 5) There is far too much of an emphasis and criticism about FDG. While it is valuable for providing some context, it is referenced too often and tends to distract from the focus of the study. FDG is not intended to demonstrate the metabolic fate of glucose, whether it is combined with ¹¹C-acetate or not. Consider revising this emphasis, as it is quite distracting (unnecessarily) - the data is good enough to stand on its own without the comparison.
- 6) Lines 277-279 - the authors state 'enhanced fatty acids metabolism in BAT upon cold exposure was observed by the ¹⁸FTHA -PET imaging in both rodent models and humans (37)' - it is

important to note that FTHA reflects circulating FA uptake, but BAT thermogenesis appears primarily fueled by intracellular fatty acids.

Reviewer 1:

Major: 1. Based on the current results, glucose is metalized rather than oxidized. One possibility is that glucose is metabolized for *de novo* fatty acids synthesis. To provide evidence that glucose is oxidized, CO₂ need to be collected and analyzed its carbon labeling.

Response: We agree that measuring the enrichment of ¹³CO₂ from the [U-¹³C]glucose tracer is the best assay to directly show glucose oxidation, as CO₂ is the direct product of glucose oxidation. We found that the relative CO₂ enrichment in plasma was significantly induced after 10 days of cold exposure, which was included as Fig.2H in the revised manuscript.

2. Injection of MPC inhibitors into mice was a whole-body effect. MPC inhibitors target every organ. Therefore, the body temperature phenotype could not be considered as a direct consequence of impaired BAT thermogenesis. Furthermore, the authors need to demonstrate whether the thermogenesis function of brown fat is affected by MPC inhibition by monitoring brown fat temperature or brown fat cell thermogenesis.

Response: We agree that IP injection of MPC inhibitors targets multiple organs of the mice. Our glucose tracing data showed that MPC inhibitor repressed the enrichment of TCA cycle intermediates in all adipose tissues, including BAT, sWAT and gWAT. We used an infrared camera to monitor the body temperature of CHC-treated mice. We found the overall body temperature, not only BAT, was lower after CHC treatment. We appreciated your advice on direct BAT temperature measurements, but we don't have access to those methods in our institute yet. We will try to find collaborators to complete those measurement in future studies.

In the meantime, we discussed that “a currently preprinted study showed that mice with BAT-selective deletion of MPC1 (BAT-MPC1) also had significantly lower core body temperatures than their littermate controls”. One of our authors, Dr. Jared Rutter, is also an author on that manuscript of BAT-MPC1 mice. Dr. Rutter knew both stories since January 2019. We decided to use chemical inhibition in our study, as Dr. Villanueva was working on the genetic mice model. Our independent studies showed that both BAT-MPC1-KO and MPC inhibitor impaired glucose metabolism in BAT and body temperature control in mice.

We agree that, with our own data using the MPC inhibitor, “the body temperature phenotype could not be considered as a direct consequence of impaired BAT thermogenesis.” We therefore have toned down the conclusion of MPC on BAT thermogenesis, and we focused on the cold-induced glucose oxidation in BAT in the revised manuscript. We changed our manuscript title to “Cold exposure enhances glucose oxidation in brown adipose tissue”.

3. How about the labeling information of other metabolites such as G3P, fatty acids? Does the glucose only function to fuel the TCA cycle in brown fat?

Response: Thanks for the advice to check the enrichment of G3P and fatty acids. Cold exposure significantly induced the enrichment of m+3 G3P, although the G3P level was reduced upon cold exposure. These data suggested that, in addition to fueling the TCA cycle, glucose can also provide G3P for TG synthesis.

Cold exposure also significantly induced the enrichment of fatty acids (palmitate), but the levels of enrichment are very low (less than 0.5%) in palmitate in both groups. These data suggested the activity of *de novo* fatty acids synthesis was very low in our

experiment setting. Together with the high enrichment of TCA cycle intermediates, our data suggest glucose is primarily used for oxidation and TG synthesis, and cold exposure increases both glucose-dependent metabolic pathways.

4. As fatty acids are believed to act as the main fuels for brown fat thermogenesis, it is therefore of interest to discuss why brown fat mitochondria metabolize glucose to fuel the TCA cycle in parallel. The authors should discuss possible reasons in the discussion part.

Response: Thanks for your advice. Our glucose tracing showed that upon chronic 10 days cold exposure, “brown fat mitochondria metabolize glucose to fuel TCA cycle”. In comparison, most of the earlier studies (PMID: 3730946, PMID: 20124047, PMID: 28089568 and PMID: 25681456) showed that, upon acute cold exposure, “fatty acids are believed to act as the main fuels for brown fat thermogenesis”. One of these studies (PMID: 20124047) shows that the total lipid content of BAT is significantly reduced after cold exposure. It is likely that, when intracellular TG is reduced after chronic cold exposure, glucose acts as another fuel for brown fat thermogenesis. We included further discussion in the revised manuscript.

Minor: 1. How enrichment is calculated? How about total abundance? These information need to be provided.

Response: Thanks for the advice. We will add the calculation in the method section of the revised manuscript.

To determine the relative metabolite abundance across samples, the area of the total ion current peak for the metabolite of interest was normalized to protein content. The mass isotopomer distribution analysis measured the fraction of each metabolite pool that contained every possible number of ^{13}C atoms: a metabolite could contain 0, 1, 2, ...n ^{13}C atoms, where n = the number of carbons in the metabolite. For each metabolite, an informative fragment ion containing all carbons in the parent molecule was analyzed by MATLAB software (MathWorks, CA, USA). The abundance of all mass isotopomers was integrated from m+0 to m+n, where m = the mass of the fragment ion without any ^{13}C . The abundance of each mass isotopomer was then corrected mathematically to account for natural abundance isotopes and finally converted into a percentage of the total pool.

2. What's the N number for cell culture experiments? n=3 is 3 biological experiments or technical replicates?

Response: All the cell culture experiments were repeated for at least 3 times, and the data were shown as 3 biological repeats in one representative experiment.

3. Brown fat uptakes glucose upon acute activation. The authors explored the metabolic fate of glucose in 10-days cold-acclimated mice. How about the acute activation condition? Is the metabolic fate of glucose different from cold-acclimated mice?

Response: A recent study showed that cold exposure increased succinate in BAT. This study also performed the glucose tracing assays in mice upon acute (3 hours) cold exposure, and there were no changes in the enrichment of glycolytic and TCA cycle intermediates. It is likely that, when intracellular TG is reduced after chronic 10 days cold exposure, glucose acts as another fuel for brown fat thermogenesis. We further discuss the difference between acute and chronic cold exposure in the discussion section of the revised manuscript.

4. In fig 6c, it seems that the body temperatures of MPC blocked mice were recovered gradually, which indicates compensatory adaptation. Could the authors please comment on this point?

Response: There are two potential compensatory adaptations, upon MPC inhibition. Mitochondria can uptake 3 carbon units through pyruvate-alanine cycling, which has been reported in the liver-specific MPC mice (PMID: 26344101). Blocking MPC-mediated glucose oxidation can also be compensated by fatty acid oxidation, which is independent of MPC. We further discussed this in the revised manuscript.

5 Line 486, differentiated should be Differentiated.

Response: Thanks. We made the change in the revised manuscript.

Reviewer 3:

Critique:

This is a very elegantly performed study that is certainly very timely for the field, well written and good description of methods. It is nice to see experiments performed in male and female mice and data presented disaggregated. The metabolomics approaches used provide valuable insight into the oxidative fate of glucose in both unstimulated and stimulated BAT, 'browned/beiged' sWAT and gWAT. The oxidative fate of glucose in BAT is of particular interest and the results gained will prove invaluable not only for future studies applying similar methods, but also in the interpretation of results generated by FDG PET or in the development of other PET radiotracers interested in examining BAT oxidative metabolism. There are, however, several critical gaps that warrant further clarification, some of which can likely be addressed through additional experiments.

Major

1a) The in vivo measures of glucose flux in BAT and its subsequent metabolic fate provides a valuable extension of the currently available data derived in vitro. However, there are two critical gaps that could assist in completing the story. First, the lack of whole-body CO₂ production, expired ¹³CO₂ and tissue ¹³CO₂ leaves a significant gap and therefore leads to several unverified assumptions within the study. If interested in determining the rate of oxidation of ¹³C-labeled glucose, you must at least determine the rate of ¹³CO₂ produced either at the whole-body and/or by the tissue. It is difficult to investigate the oxidation of a substrate without knowing the rate with which the final downstream product (CO₂) is produced. Several inferences could be made with the measure of whole-body VCO₂ and ¹³CO₂. Both are also simple measures to make with any metabolic system. If you assume a large proportion of the increased thermogenesis after 10 d of cold exposure is due to BAT, then several inferences could be made about ¹³CO₂/totalCO₂ production and its relationship to BAT glucose oxidation. Fick-derived measures of tissue ¹³CO₂ would be helpful too, but more difficult to execute. The addition of indirect calorimetry, to quantify total VCO₂ and substrate utilisation, combined with measures of ¹³C carbon dioxide enrichment (¹³CO₂) would provide valuable context to the current data.

Response: Thanks for your advice. The other reviewer (**major 1**) also suggested us to measure CO₂ enrichment. Please see our response on page 1.

1b) The second gap is in examining the non-oxidative fate of the glucose (de novo fatty acid synthesis, glyceroneogenesis). Several lines of evidence from in vitro experiments (PMID: 27110487), inferences made from cold-acclimation-induced changes in gene expression (PMID: 25681456) and in vivo studies (PMID: 15688247, 10198378) suggest that a large proportion of glucose taken up by BAT under stimulated conditions is used for de novo fatty acid synthesis and glyceroneogenesis, to support the high intracellular TG turnover. The reductionist perspective presented here omits this critical fate of glucose. This manuscript would benefit from examining at least the enrichment of glycerol-3-phosphate/glycerol and FA isotopologues in BAT to provide valuable context.

Response: Thanks for your advice. The other reviewer (**major 3**) also suggested examining the enrichment of glycerol-3-phosphate/glycerol and FA isotopologues. Please see our response on page 1.

2a) The conclusion that glucose oxidation is required for thermogenesis in the cold-activated brown adipose tissue is a significant overstatement of the study findings and not fully supported by the data provided. Glucose oxidation in BAT may indeed increase in the cold, but the evidence supporting that it is required for thermogenesis is lacking. The fact that T_{core} (Figure 6C) appears to recover over time following MPC inhibition, which is the only data available to support this premise, seems to refute that conclusion (shows the effect is transient).

Response: We agree that glucose oxidation is required for thermogenesis in the cold-activated brown adipose tissue is an overstatement, and the other reviewer (**major 2**) also had a similar question. Please see our response on page 1.

The other reviewer (Minor 4) also had a question about the recovery of body temperature after the injection of an MPC inhibitor. Please see our response on page 2.

2b) As total energy expenditure increases ~5-fold (again, could be measured by indirect calorimetry) or more at such an intense cold, does it not make sense that glucose-derived TCA cycle intermediates also increase? Quantitatively, this may not be all that significant as iTG is the predominant fuel source. Fig 4 suggests that the cold-induced increase in glucose oxidation likely accounts for ~10% of total carbon flux through TCA cycle (citrate m+0 went down ~10% and m+3 went up ~10%). What both the in vivo and in vitro data suggests is that a certain proportion of glucose taken up by the tissue is directed towards oxidation and that this flux increases in the cold or with CL-treatment. It could be worth considering Seahorse experiments to examine the effects of inhibiting MPC (and PDH) on OCR.

Response: We agree that TG has been accepted as the predominant fuel source in BAT, but mice defective in brown fat lipolysis are not cold sensitive (PMID: 28988822). Dr. Ronald Kahn's recent study also showed that BAT had the flexibility in glucose and fat acids utilization (PMID: 31000437). Our semi-quantitative tracing data showed that, 15min after [^{13}C]glucose injection, about 40% of the TCA cycle intermediates were enriched in BAT upon chronic cold exposure (Appendix Figure S1D). These data suggested BAT can also use glucose as another major fuel source. One possibility is that the kinetics of glucose and TG utilization are different in BAT upon cold exposure. Although TG is the predominant fuel source upon acute cold exposure, after chronic cold exposure, BAT can also use the available glucose after the injection of [^{13}C]glucose tracer.

Fig. 4 compared the glucose metabolism between differentiated brown adipocytes and the undifferentiated SVF, and Fig. 5 showed the effect on glucose metabolism in CL-activated brown adipocytes. We performed seahorse experiments to measure the effect on UK5099 and CHC on oxygen consumption rate (OCR). Similar to our in vitro [^{13}C]glucose tracing, both UK5099 and CHC significantly reduced OCR in primary brown adipocytes (Fig.5A).

Minor

1) In Fig. 6, it would be useful to have the enrichment of the glycolytic metabolites (upstream of the inhibition).

Response: Thanks for the advice. Both CHC and UK5099 treatment didn't change the enrichment of glycolytic intermediates in vitro, but CHC repressed the enrichment of glycolytic intermediate in BAT in vivo. These data are included as Fig. 5 in the revised manuscript.

2) Figures 2 and 3, would benefit from presenting tissue enrichments of ^{13}C -glucose.

Response: Thanks for the advice. The data of tissue enrichments of ^{13}C -glucose are added to the revised manuscript.

3) On Line 244 - the authors report that '...over 40% of the TCA cycle intermediates in BAT was labeled upon a bolus IP injection of [^{13}C]glucose tracer, our results suggested that glucose served as an important energy source for mitochondrial oxidation in the BAT of mice upon chronic cold exposure.' It is unclear where this 40% is derived from. Figure 1E shows 20%?

Response: Appendix Figure S1D showed that m+0 (unlabeled) fraction of most TCA cycle intermediates was less than 60% in BAT upon cold exposure, which suggested over 40% was labeled

as m+1,2...n from [U-¹³C]glucose tracer.

4) Fig 6. seems to also suggest that there may be effects on anaplerotic pathways - has this been examined?

Response: Thanks for the advice. The m3 enrichment of TCA cycle intermediates represents the glucose-dependent anaplerotic reaction, pyruvate carboxylase (PC). MPC inhibition decreased both glucose-dependent oxidation (PDH) and anaplerosis (PC). This is not surprising, as both PDH and PC use mitochondrial pyruvate as substrate, and MPC inhibition impairs mitochondrial pyruvate uptake. These data are included as Fig. 5 in the revised manuscript.

5) There is far too much of an emphasis and criticism about FDG. While it is valuable for providing some context, it is referenced too often and tends to distract from the focus of the study. FDG is not intended to demonstrate the metabolic fate of glucose, whether it is combined with ¹¹C-acetate or not. Consider revising this emphasis, as it is quite distracting (unnecessarily) - the data is good enough to stand on its own without the comparison.

Response: Thanks for your positive comment on our tracing approach in this study. We removed most of the discussion and comparison to FDG in the revised manuscript.

6) Lines 277-279 - the authors state 'enhanced fatty acids metabolism in BAT upon cold exposure was observed by the ¹⁸FTHA -PET imaging in both rodent models and humans (37)' - it is important to note that FTHA reflects circulating FA uptake, but BAT thermogenesis appears primarily fueled by intracellular fatty acids.

Response: Thanks for the advice. We agree that "FTHA reflects circulating FA uptake" not fatty acids oxidation. Our glucose tracing showed that upon chronic 10 days cold exposure, brown fat mitochondria metabolized glucose to fuel the TCA cycle. We would like to use a similar stable isotope tracing approach to quantify the contribution of extracellular fatty acids to BAT thermogenesis. Several earlier studies (PMID: 3730946, PMID: 20124047, PMID: 28089568 and PMID: 25681456) showed that, upon acute cold exposure, "BAT thermogenesis appears primarily fueled by intracellular fatty acids". One of these studies (PMID: 25681456) showed that the 21 days cold-induced increase in oxidative activity was meaningfully blunted by nicotinic acid, a lipolysis inhibitor, and it worth to note that lipolysis inhibitor treatment also significantly decreased glucose uptake and fatty acid uptake in BAT. We included further discussion in the revision.

Dear Lei,

Thank you for submitting the revised version of your manuscript. It has now been seen by both of the original referees.

As you can see, the referees find that the study is significantly improved during revision. However, there are a couple of outstanding issues that need to be addressed prior to publication. Please make sure that all textual changes are visible in the document by keeping 'track changes' on.

1. As I mentioned in my earlier email from 24.04.2020, referee 3 does not find the infrared camera to be a reliable method for measurement of tissue thermogenesis/temperature. Therefore please keep the data in, but please discuss the caveats of this method as explained by referee #3 in his/her point 4 and tone down the conclusions regarding the effects of MPC1 inhibition on BAT thermogenesis.

2. Both referees find that the abstract does not fully reflect the findings in the manuscript (referee #1, referee #3 point 1).

3. Referee #3 finds that the G3P enrichment experiment in BAT that was requested during the first review round should also be demonstrated in female mice (point 3). Moreover, referee #3 requests the demonstration of G3P also in sWAT (point 2). As these points were not requested in the first review round, addressing them experimentally is not prerequisite for publication. However, if you have data in hand already, this would of course strengthen the manuscript.

4. Please address the remaining minor concerns of referees.

Additionally, I need you to address the below editorial points.

- Please clarify in the text and the figure legend that the data in Fig 2F and Fig S2B are calculated from the same data set.
- As per our guidelines, please add a 'Data Availability Section'. If inapplicable, please add the section anyway where you state that no data were deposited in a public database.
- Please provide 3-5 keywords for your study. These will be visible in the html version of the paper and on PubMed and will help increase the discoverability of your work.
- We noted that there are still callouts to Supplementary tables. Please update the callouts to the tables.
- We noted that the page numbers of Table of Contents of the Appendix is currently missing.
- Papers published in EMBO Reports include a 'Synopsis' to further enhance discoverability.

Synopses are displayed on the html version of the paper and are freely accessible to all readers. The synopsis includes a short standfirst summarizing the study in 1 or 2 sentences that summarize the key findings of the paper and are provided by the authors and streamlined by the handling editor. I would therefore ask you to include your synopsis blurb.

- In addition, please provide an image for the synopsis. This image should provide a rapid overview of the question addressed in the study but still needs to be kept fairly modest since the image size cannot exceed 550x400 pixels.
- Our production/data editors have asked you to clarify several points in the figure legends (see attached document). Please incorporate these changes in the attached word document and return it with track changes activated.

Thank you again for giving us to consider your manuscript for EMBO Reports, I look forward to your minor revision.

Kind regards,

Deniz

--

Deniz Senyilmaz Tiebe, PhD
Editor
EMBO Reports

Referee #1:

This is an excellent revision even that we are currently in a global pandemic that may preclude the completion of these studies in a timely manner. The authors have thoroughly responded to the reviewers' comments, and added some new experiments that are quite impressive. These significantly strengthen their conclusion. As I looking at the revised manuscript, I appreciated the authors highlighting their chronic cold exposure data and differentiated them from previously published acute cold exposure data. Therefore, I would recommend the authors also highlight this in their title and abstract: the paper was talking about chronic cold exposure. By reasoning this way, I suddenly realized that their in vitro data were nevertheless acute CL stimulation, representing/mimicking acute cold exposure. Thus, there is an inconsistency. Moreover, I guess in all their seahorse data n=6-7 are technical repeats rather than biological repeats. The authors need to clarify these points.

Referee #3:

The authors should be commended for the additional work performed to strengthen the manuscript and to respond to some of the critiques. However, some of the responses lacked clarity and it was not evident, where, if any, changes were made to the text or figures to respond to the reviewer's critiques (please use underlined/red text or something else to distinguish what is new). The authors directed the reviewer to responses given to the other reviewer, but often those questions and responses were unrelated or inadequately responded to the critique. There are several outstanding questions that require careful consideration.

1) The title and abstract do not fully reflect the findings of this paper. The key findings of this study are, as stated in line 126-127, that 'Together with the high enrichment of TCA cycle intermediates, our data suggest cold exposure increases glucose-dependent oxidation and TG synthesis (namely glyceroneogenesis) in BAT.' The study in its current form is still focused on the glucose oxidation, and gives the impression that this is the principal and most significant fate of glucose in chronically cold-exposed mice and diminishes the non-oxidative fate of glucose (through glyceroneogenesis).

2) With regards to the browning, it would have been beneficial to also demonstrate the enrichment of G3P in sWAT. Presumably, if there is an increased browning, and expect an accompanied increase in thermogenic activity, both glucose oxidation and glyceroneogenesis (increased glucose-derived G3P) should be increased (like BAT).

3) Is G3P enrichment also increased in the BAT of female mice? Similar or different than the male mice?

4) In response to the effect of MPC1 inhibition on the body temperature maintenance of the mice upon cold exposure, there is an inadequate explanation for this effect. Firstly, thermal imaging is an unreliable measure of anything more than surface temperature. It does not detect BAT temperature or body temperature, only surface temperature (by its very function, these cameras measure the emissivity of the radiation emitted by a surface). Secondly, reference to a pre-print using a BAT-selective deletion of MPC1 is also irrelevant. The present model is not a BAT-selective deletion and the pre-print has not been reviewed yet. With the absence of MPC1, couldn't FAO be increased to compensate for the absence of pyruvate transport (as suggested on lines 312-313)? Couldn't amino acids compensate for the absence of the anaploretic reactions that pyruvate might be involved in? If it were only relevant for BAT thermogenesis, couldn't skeletal muscle increase their shivering activity to compensate, or does the MPC1 inhibition also impair skeletal muscle metabolism? Could the MPC inhibition also be having effects at the liver - perhaps reducing hepatic glucose output, which would decrease circulating glucose which could suppress both shivering and BAT through centrally-mediated mechanisms?

The statement 'Together, our results indicate that mitochondrial pyruvate uptake and oxidation is required for optimal thermogenesis in BAT and beige adipose tissue' is an over-extrapolation of findings - there is no measure of BAT or beige thermogenesis in the present study. As commented above, it is unclear why body temperature is undefended following CHC injection. While the OCR measures are interesting, the most that could be inferred from these results is that MPC1 inhibition reduces brown adipocyte respiration.

5) The manuscript in its present form is still filled with comparisons to FDG. This comparison has no relevance to this study. Users of FDG know that it doesn't measure the metabolic fate of glucose. If they were interested in the tissue metabolic fate of glucose they could use the approach used here or ¹¹C-glucose. As previously suggested, reference to FDG should be removed. Further, lines 254 - 256 are not accurate. FDG PET with ¹¹C-acetate does not assist in determining the fate of glucose. They are completely independent measures that represent two distinctly different outcomes - FDG = glucose uptake only, ¹¹C-acetate = TOTAL TCA cycle flux (independent of the source of acetyl-coA)

6) The manuscript would benefit from a careful revision of the use of references. There are several instances where there is an inaccurate depiction of study methods or findings including, but certainly not limited to, the following:

ex.1: lines 107-108 and line 316 - the authors use 3 different references regarding the use of ¹⁸FTHA -PET imaging in both rodent models and humans. All three are human only studies. Others have used FTHA in rodents, but these are not them.

ex.2: As stated by a previous reviewer, the statement that 'it remains unclear whether the thermogenic brown/beige adipose tissue could completely oxidize glucose in vivo' and citing the work of Hankir and Klingenspor is a bit misleading, as there is work discussed within this review as well as the work that's been done in vivo by others (eg. Ma & Foster and more recently PMID: 32769145).

ex.3: Line 36 - Cypess et al, 2013, does not report uncoupled respiration, only Basal.

7) This manuscript would benefit from a detailed limitations section, of which, many of been referenced by previous reviewers. Namely, one must consider the single time point of measurement and how this truly reflects glucose kinetics under steady state conditions. One must also consider whether there are any isotopic effects resulting from the competition between endogenous and exogenous (labelled) glucose, and whether the dose of ¹³C-glucose given increases glycemia. Based on study of Nagao et al. referenced here which uses half the dose given in the present study, it will definitely increase glycemia by a significant extent, which will impact BAT glucose uptake and ultimately its fate, which may explain some of the outcomes presented here. Would the same outcomes be expected if only trace amounts of glucose were given?

Referee #1:

This is an excellent revision even that we are currently in a global pandemic that may preclude the completion of these studies in a timely manner. The authors have thoroughly responded to the reviewers' comments, and added some new experiments that are quite impressive. These significantly strengthen their conclusion. As I looking at the revised manuscript, I appreciated the authors highlighting their chronic cold exposure data and differentiated them from previously published acute cold exposure data. Therefore, I would recommend the authors also highlight this in their title and abstract: the paper was talking about chronic cold exposure. By reasoning this way, i suddenly realized that their *in vitro* data were nevertheless acute CL stimulation, representing/mimicking acute cold exposure. Thus, there is an inconsistency. Moreover, i guess in all their seahorse data n=6-7 are technical repeats rather than biological repeats. The authors need to clarify these points.

We thank for your advice.

We added “chronic cold exposure” to the title, and we also updated the abstract and text.

An earlier study showed that 6 hours CL treatment induces the expression of glucose metabolic enzymes in brown adipocytes (PMID: 25516548). In this study, we performed *in vitro* tracing in adipocytes pre-treated with CL for 4 hours, followed by 2 hours tracing with CL treatment. We agree that longer (24 hours) CL pre-treatment will be a better mimic of chronic cold exposure, and we will test this in future studies.

The seahorse experiment was performed with 6-7 wells of adipocytes, which were plated before the 8 days differentiation process. We believe that they are biological repeats.

Referee #3:

The authors should be commended for the additional work performed to strengthen the manuscript and to respond to some of the critiques. However, some of the responses lacked clarity and it was not evident, where, if any, changes were made to the text or figures to respond to the reviewer's critiques (please use underlined/red text or something else to distinguish what is new). The authors directed the reviewer to responses given to the other reviewer, but often those questions and responses were unrelated or inadequately responded to the critique. There are several outstanding questions that require careful consideration.

We thank for your advice, and we are glad our previous revision was strengthened with additional experiments. In the first round of revision, a world file with tracing was uploaded. As we made significant changes to our manuscript, we removed the tracing in merged pdf file. We marked all the changes in red with comments and tracking in this round of revision.

1) The title and abstract do not fully reflect the findings of this paper. The key findings of this study are, as stated in line 126-127, that 'Together with the high enrichment of TCA cycle intermediates, our data suggest cold exposure increases glucose-dependent oxidation and TG synthesis (namely glyceroneogenesis) in BAT.' The study in its current form is still focused on the glucose oxidation, and gives the impression that this is the principal and most significant fate of glucose in chronically cold-exposed mice and diminishes the non-oxidative fate of glucose (through glyceroneogenesis).

We agree that our data suggest cold exposure increases glucose-dependent oxidation and glyceroneogenesis in BAT. Our *in vivo* tracing data showed that, upon chronic cold exposure, G3P enrichment is higher in the BAT and sWAT of male mice, and BAT, sWAT and gWAT of female mice, upon chronic cold exposure. Consistent with the previous report (PMID: 29934543), our *in vitro* tracing data also show that G3P enrichment is higher in CL-treated brown adipocytes, but the relative abundance of m+3 G3P is lower after CL treatment. Furthermore, MPC inhibitors decrease OCR and the enrichment of TCA cycle intermediates in the cultured brown adipocytes, without altering G3P enrichment or abundance. Since the glucose-dependent oxidation showed similar induction in both *in vivo* and *in vitro* models, our current study focuses on the glucose oxidation. We agree that glyceroneogenesis is important in BAT, and we will further look into it in the future studies.

2) With regards to the browning, it would have been beneficial to also demonstrate the enrichment of G3P in sWAT. Presumably, if there is an increased browning, and expect an accompanied increase in thermogenic activity, both glucose oxidation and glyceroneogenesis (increased glucose-derived G3P) should be increased (like BAT).

3) Is G3P enrichment also increased in the BAT of female mice? Similar or different than the male mice?

We thank for your advice. Similar to the induction in male mice, chronic cold exposure also increased the enrichment of m+3 G3P in the BAT of female mice. The induction of enrichment of m+3 G3P were also detected in the sWAT of both female and male mice, as well as the gWAT of female mice. These data are included as figure EV.3G-H in the revised manuscript.

4) In response to the effect of MPC1 inhibition on the body temperature maintenance of the mice upon cold exposure, there is an inadequate explanation for this effect. Firstly, thermal imaging is an unreliable measure of anything more than surface temperature. It does not detect BAT temperature or body temperature, only surface temperature (by its very function, these cameras measure the emissivity of the radiation emitted by a surface). Secondly, reference to a pre-print using a BAT-selective deletion of MPC1 is also irrelevant. The present model is not a BAT-selective deletion and the pre-print has not been reviewed yet. With the absence of MPC1, couldn't FAO be increased to compensate for the absence of pyruvate transport (as suggested on lines 312-313)? Couldn't amino acids compensate for the absence of the anaplerotic reactions that pyruvate might be involved in? If it were only relevant for BAT thermogenesis, couldn't skeletal muscle increase their shivering activity to compensate, or does the MPC1 inhibition also impair skeletal muscle metabolism? Could the MPC inhibition also be having effects at the liver

- perhaps reducing hepatic glucose output, which would decrease circulating glucose which could suppress both shivering and BAT through centrally-mediated mechanisms?

The statement 'Together, our results indicate that mitochondrial pyruvate uptake and oxidation is required for optimal thermogenesis in BAT and beige adipose tissue' is an over-extrapolation of findings - there is no measure of BAT or beige thermogenesis in the present study. As commented above, it is unclear why body temperature is undefended following CHC injection. While the OCR measures are interesting, the most that could be inferred from these results is that MPC1 inhibition reduces brown adipocyte respiration.

We agree that thermal imaging is an unreliable measure of body temperature, and we didn't try to include the infrared image in the previous revision. We only took images of one pair of mice at different time points, which was only included in our previous point-to-point letter, not in the manuscript. Secondly, the earlier pre-print manuscript about BAT-selective deletion of MPC1 is now accepted in eLife (eLife 2020;9:e52558), which showed that conditional deletion of Mpc1 in brown adipocytes leads to the impaired cold adaptation.

We agree that our study didn't directly measure the BAT or beige thermogenesis, and the CHC injection was not sufficient to show the role of BAT MPC in body temperature maintenance. We removed all the conclusion about thermogenesis, and we changed the statement to "mitochondrial pyruvate uptake and oxidation serves an important energy source in the chronic cold exposure activated BAT and beige adipose tissue, which supports a role for glucose oxidation in brown fat thermogenesis." Line 34-36, 75-76, 301-303.

5) The manuscript in its present form is still filled with comparisons to FDG. This comparison has no relevance to this study. Users of FDG know that it doesn't measure the metabolic fate of glucose. If they were interested in the tissue metabolic fate of glucose they could use the approach used here or ^{11}C -glucose. As previously suggested, reference to FDG should be removed. Further, lines 254 - 256 are not accurate. FDG PET with ^{11}C -acetate does not assist in determining the fate of glucose. They are completely independent measures that represent two distinctly different outcomes - FDG = glucose uptake only, ^{11}C -acetate = TOTAL TCA cycle flux (independent of the source of acetyl-coA)

We thank for your comment that $[\text{U-}^{13}\text{C}]$ glucose tracing is different from the FDG-PET, and we appreciate that you find tracing studies provides new information to BAT metabolism. The comparisons to FDG were added after our previous submissions, since many other readers might not be fully clear about the differences between $[\text{U-}^{13}\text{C}]$ glucose tracing and FDG-PET. As you advised in the first revision, we removed the criticism about FDG. We kept some comparison to FDG, since we feel it might help the wide reader of this journal.

One example is that 'FDG = glucose uptake only, ^{11}C -acetate = TOTAL TCA cycle flux (independent of the source of acetyl-coA)'. Previous study used ^{11}C -acetate tracing method to conclude the activated oxidative metabolism in BAT after cold stimulation (PMID: 28134339). To our knowledge, ^{11}C -acetate-derived acetyl-CoA can be oxidized by TCA cycle or used for fatty acids synthesis. We changed lines 254-56 to ' ^{18}F -FDG-PET and ^{11}C -acetate-PET indirectly indicated the complete glucose oxidation in BAT *in vivo*'.

6) The manuscript would benefit from a careful revision of the use of references. There are several instances where there is an inaccurate depiction of study methods or findings including, but certainly not limited to, the following:

ex.1: lines 107-108 and line 316 - the authors use 3 different references regarding the use of ¹⁸FTHA -PET imaging in both rodent models and humans. All three are human only studies. Others have used FTHA in rodents, but these are not them.

ex.2: As stated by a previous reviewer, the statement that 'it remains unclear whether the thermogenic brown/beige adipose tissue could completely oxidize glucose in vivo' and citing the work of Hankir and Klingenspor is a bit misleading, as there is work discussed within this review as well as the work that's been done in vivo by others (eg. Ma & Foster and more recently PMID: 32769145).

ex.3: Line 36 - Cypess et al, 2013, does not report uncoupled respiration, only Basal.

We apologize for citing the inaccurate studies, and we made the following corrections.

Ex.1, we added PMID: 24423363 and PMID: 25681456 to lines 107-108.

We added PMID: 28130074 and PMID: 25681456 to line 316.

Ex. 2, we agree that early studies, including the more recently PMID: 32769145, used FDG or DOG to show the glucose uptake in the activated BAT, as they are non-metabolizable glucose tracers. Other studies, including Ma & Foster, measured plasma arteriovenous metabolic concentration differences across IBAT to show the glucose uptake and lactate/pyruvate secretion. As recent tracing study showed that lactate could also be a fuel for BAT (PMID: 32791100), we believe that the enrichment of TCA cycle intermediates from [U-¹³C]glucose tracer is a more direct analysis of glucose oxidation. With these reasons, we change the statement to 'it is still not fully clear whether the thermogenic brown/beige adipose tissue could completely oxidize glucose in vivo', lines 24, 62-63

Ex. 3, the citation of 'Cypess et al, 2013' is replaced with Chouchani et al, 2016.

We also made these corrections. Barquissau et al, 2016; Held et al., 2018 are removed from line 123. Kory et al, 2018 is removed from line 265. Jiang et al, 2017 is removed from line 253.

7) This manuscript would benefit from a detailed limitations section, of which, many of been referenced by previous reviewers. Namely, one must consider the single time point of measurement and how this truly reflects glucose kinetics under steady state conditions. One must also consider whether there are any isotopic effects resulting from the competition between endogenous and exogenous (labelled) glucose, and whether the dose of ¹³C-glucose given increases glycemia. Based on study of Nagao et al. referenced here which uses half the dose given in the present study, it will definitely increase glycemia by a significant extent, which will impact BAT glucose uptake and ultimately its fate, which may explain some of the outcomes presented here. Would the same outcomes be expected if only trace amounts of glucose were given?

We agree that a detailed limitations section helps this study, and we added a paragraph to the discussion section. We also discussed the 'competition between endogenous and exogenous (labelled) glucose' in the results section, lines 151-154.

Dear Lei,

Thank you for submitting your revised manuscript. I have now looked at everything and all is fine. Therefore I am very pleased to accept your manuscript for publication in EMBO Reports.

Congratulations on a nice study!

Kind regards,

Deniz

--

Deniz Senyilmaz Tiebe, PhD
Editor
EMBO Reports

--

At the end of this email I include important information about how to proceed. Please ensure that you take the time to read the information and complete and return the necessary forms to allow us to publish your manuscript as quickly as possible.

As part of the EMBO publication's Transparent Editorial Process, EMBO reports publishes online a Review Process File to accompany accepted manuscripts. As you are aware, this File will be published in conjunction with your paper and will include the referee reports, your point-by-point response and all pertinent correspondence relating to the manuscript.

If you do NOT want this File to be published, please inform the editorial office within 2 days, if you have not done so already, otherwise the File will be published by default [contact: emboreports@embo.org]. If you do opt out, the Review Process File link will point to the following statement: "No Review Process File is available with this article, as the authors have chosen not to make the review process public in this case."

Should you be planning a Press Release on your article, please get in contact with emboreports@wiley.com as early as possible, in order to coordinate publication and release dates.

Thank you again for your contribution to EMBO reports and congratulations on a successful publication. Please consider us again in the future for your most exciting work.

THINGS TO DO NOW:

You will receive proofs by e-mail approximately 2-3 weeks after all relevant files have been sent to our Production Office; you should return your corrections within 2 days of receiving the proofs.

Please inform us if there is likely to be any difficulty in reaching you at the above address at that

time. Failure to meet our deadlines may result in a delay of publication, or publication without your corrections.

All further communications concerning your paper should quote reference number EMBOR-2020-50085V3 and be addressed to emboreports@wiley.com.

Should you be planning a Press Release on your article, please get in contact with emboreports@wiley.com as early as possible, in order to coordinate publication and release dates.

Corresponding Author Name: Lei Jiang

Manuscript Number: EMBOR-2020-50085V1